# Microstructure Observations and Mixing Parameterizations along an Atlantic Transect in Very Weak Turbulence

Niek Kusters[1], Sjoerd Groeskamp[1], Bieito Fernandez Castro[2], and Hans van Haren[1]

[1]NIOZ Royal Netherlands Institute for Sea Research, Texel, The Netherlands
[2]Ocean and Earth Science, National Oceanography Centre, University of Southampton, Southampton, United Kingdom

**Correspondence:** Niek Kusters (niek.kusters@nioz.nl)

**Abstract.** Microstructure measurements of shear and temperature can be used to calculate ocean turbulent dissipation rates and diffusivities. Here microstructure observations are taken along an transect in the North Atlantic, that includes observations of very weak deep ocean turbulence. In this paper we show the necessity of using the thermistor probes, instead of the more common shear probes, to calculate dissipation rates when they are smaller than $1 \times 10^{-10}$ W kg$^{-1}$. Profiles of combined dissipation rates from the shear and thermistor probes are then compared to the finescale strain parameterization and Thorpe sorting method. Based on this comparison, recommendations and restrictions are suggested for applying both parameterizations in a weakly turbulent environment. The results indicate that temperature-based strain provides improved estimates of dissipation rates in the deep ocean where density gradients are small, while density-based strain provides better results otherwise. We find that Thorpe based estimates are very accurate when pre-existing knowledge of the turbulent kinetic energy dissipation rate $\varepsilon$ is used. When this knowledge is not available, using climatological mean estimates of $\varepsilon$ can allow for more detailed estimates of dissipation by applying the Thorpe resorting method. Finally, we employ the triple decomposition framework to get more insights in the relative roles of dianeutral and isoneutral mixing processes, and use this to calculate the dianeutral and isoneutral diffusivities. It turns out that the triple decomposition is generally not a good predictor of the isoneutral diffusivity. Overall, this paper has assessed the potential of direct observations and parameterizations of dissipation and showed that dissipation rates can be estimated quite well within a factor 5 between different methods, but it becomes difficult to achieve higher accuracy.

## 1 Introduction

Dianeutral mixing –mixing across layers of different water density– is caused by a wide variety of processes, such as winds, tides and geostrophic currents (MacKinnon et al., 2017; Whalen et al., 2020). It plays an important role in circulation (de Lavergne et al., 2022) –involved in the closure of the Meridional Overturning Circulation (Wunsch and Ferrari, 2004; Melet et al., 2022; Cimoli et al., 2023)–, the transport and distribution of tracers in the ocean such as heat, carbon (Tatebe et al., 2018) or contaminants and nutrients (Friedrich et al., 2011; Spingys et al., 2021), eventually impacting regional and global climate (Pradal and Gnanadesikan, 2014). Away from surface and bottom boundary layers, in the ocean interior, mixing is sustained by a field of internal waves (MacKinnon et al., 2017). The breaking of these internal waves sets off an energy cascade down to the smaller scales that eventually leads to irreversible energy dissipation and turbulent tracer mixing. The rate of energy

dissipation can be recalculated into a mixing strength or diffusivity (Osborn, 1980), which can subsequently be used to study the role of mixing in the ocean and climate system through for example numerical simulations (Melet et al., 2013).

Observational studies have shown that the dianeutral mixing strength is highly variable throughout the ocean, spanning several orders of magnitude (Polzin et al., 1997; Naveira Garabato et al., 2004). Weak diffusivities of $\mathcal{O}(10^{-5})$ m$^2$/s have been observed in the ocean interior (Munk and Wunsch, 1998), while stronger mixing rates ($\mathcal{O}(10^{-3})$ m$^2$/s) are found closer to boundaries and over rough topography (Polzin et al., 1997; Waterhouse et al., 2014; Wynne-Cattanach et al., 2024). The patchy and intermittent nature of dianeutral mixing makes it a difficult quantity to measure. Meanwhile, values found in the quiescent ocean interior can be extremely low, adding difficulty to obtaining such observations because of the high instrument precision required for measuring such low values (Scheifele et al., 2018).

Obtaining observations of the turbulent dissipation rate $\varepsilon$, and subsequently the diffusivity, requires the resolution of the observations to be $\mathcal{O}$(mm-cm) in order to resolve the velocity shear variance at the scales of the turbulent inertial subrange. This can be achieved for example with (vertical) microstructure profilers. These profilers are usually equipped with a combination of airfoil shear probes, measuring vertical velocity shear $\left(\frac{\partial u}{\partial z}\right)$, and fast-response thermistors (FP07), measuring the temperature gradient $\left(\frac{\partial T}{\partial z}\right)$. This allows for measurements of the turbulent kinetic energy (TKE) dissipation rate $\varepsilon$ and the temperature variance dissipation rate $\chi$, directly at the scales of dissipation. The specialized nature of this type of instruments cause these kind of observations to remain scarce, especially when covering the full ocean depth (Waterhouse et al., 2014). Consequently, alternative methods have been developed to indirectly estimate dissipation and mixing strengths from more readily available data such as ship-based Conductivity-Temperature-Depth (CTD) probes, moorings, and autonomous platforms (e.g., gliders or Argo floats). Some examples are the finescale parameterization (Gregg, 1989; Polzin et al., 1995, 2014), the Thorpe resorting method (Thorpe, 1977; Dillon, 1982), inverse modelling (Ganachaud and Wunsch, 2000; Sloyan and Rintoul, 2000; Zika et al., 2010; Groeskamp et al., 2017; Hautala, 2018; Kusters et al., 2024), and internal wave ray tracing (de Lavergne et al., 2020). However, the application of these parameterizations relies heavily on a variety of assumptions, adding uncertainty to estimates. The combination of observations and indirect estimates of dissipation and turbulent mixing strength, allows for calibration, fine-tuning and checking of the indirect estimates. The indirect estimates often provide possibilities to estimate mixing at spatial and temporal scales that are impossible for microscale observations. Hence combining both approaches greatly improves our understanding of turbulent mixing processes and our ability to include these effects in numerical simulations and other studies (Fox-Kemper et al., 2019).

The goals of this study are to (1) compare direct microstructure observations from both shear probes and thermistor probes with each other, and (2) compare these direct estimates to indirect estimates obtained using the finescale parameterization and the Thorpe resorting method, both derived from finescale temperature and salinity data collected with a profiler-mounted CTD, and finally (3) understand if the triple decomposition framework (Ferrari and Polzin, 2005; Naveira Garabato et al., 2016; Castro et al., 2024) can give insight for the interpretation of the results.

Various studies have reported on comparisons between microstructure observations and one or more parameterizations for the dissipation rate. These studies were mostly situated in either energetic locations such as in the Antarctic Circumpolar Current (ACC) (Thompson et al., 2007; Waterman et al., 2013; Frants et al., 2013; Park et al., 2014), in the proximity of rough

topography (Ferron et al., 1998) or in the upper parts of the ocean (Whalen et al., 2015; Howatt et al., 2021; Roget et al., 2023), i.e. locations where sensor limitations are of lesser concern. Few studies do make a comparison between direct measurements of $\varepsilon$ and one or more parameterizations in a low energetic environment, such as the Arctic Ocean (Fine et al., 2021; Baumann et al., 2023). However, these studies often only use shear-based observations of $\varepsilon$ and are thus unable to consider values below the shear noise floor. Whereas for the few studies that are concerned with thermistor-based $\varepsilon$ estimates in the deep sea (e.g., Scheifele et al. (2018); Yasuda et al. (2021)), a comparison with parameterizations is not within the scope of those studies.

This study uses data from two research cruises across the subtropical North Atlantic (Fig. 1) as part of the Mixation and Nanoplastics 2 projects. For large parts of this dataset, there is weak stratification and (very) low dissipation rates, often below the noise floor of the more conventionally used shear probes, but yet within the range of the thermistor probes. Here we assess the difference between the two types of probes on the microstructure profiler, and we compare these microstructure measurements to two parameterizations – a finescale strain parameterization and a Thorpe scale-based method–, in this dataset with weak turbulence. Additionally, we explore practical improvements to such indirect methods, e.g. the use of strain based on temperature gradients for the use in the finescale parameterization and objective criteria to exclude false overturns in the Thorpe resorting method.

The paper is structured as follows. In Section 3, the limits of both the shear and thermistor probes will be explored, showcasing the need for using thermistor probes instead of shear probes in areas of the deep sea where turbulent dissipation rates are below $1 \times 10^{-10}$ W kg$^{-1}$. In Section 4, the dissipation rates from the microstructure data are compared to the finescale-strain parameterization that uses strain based on temperature gradients and the buoyancy frequency from the CTD data. We showcase that in the deepsea, the temperature-based finescale parameterization is preferred over the more commonly used density based finescale parameterizations, due to higher instrumental noise in the salinity probe. In Section 5, the CTD-temperature data is used at smaller scales for the Thorpe resorting method. Here we find that Thorpe parameterization is able to reproduce the dissipation rates well, also in the weakly turbulent regions, but that these results are sensitive to methodological choices. We then apply the triple decomposition framework (Ferrari and Polzin, 2005; Naveira Garabato et al., 2016; Castro et al., 2024) to the collected CTD and mircostructure data (Section 6). This framework provides further insight into the relative roles of isoneutral and dianeutral processes in setting the observed T-S relations that impact the used parameterizations. These observations also allows for estimating the dianeutral diffusivities and mixing efficiency (Section 7). Finally we end with a discussion and conclusions (Section 8).

## 2  Data

In this study we use a set of microstructure profiles collected during the Mixation and Nanoplastics 2 cruises (Fig. 1). A total of 12 full-depth microstructure profiles (VMP-6000, Rockland Scientific International Inc) with standard (pumped) CTD data (SeaBird Electronics SBE 3/4/5, 64 Hz) were collected to approximately 30 meters above the seafloor. Except st. NP3, where the VMP experienced an early release approximately 700 meters above the seafloor. Additional minor problems with the FP07 thermistors were encountered at some stations, such as anomalous spikes likely caused by collisions or general malfunctioning

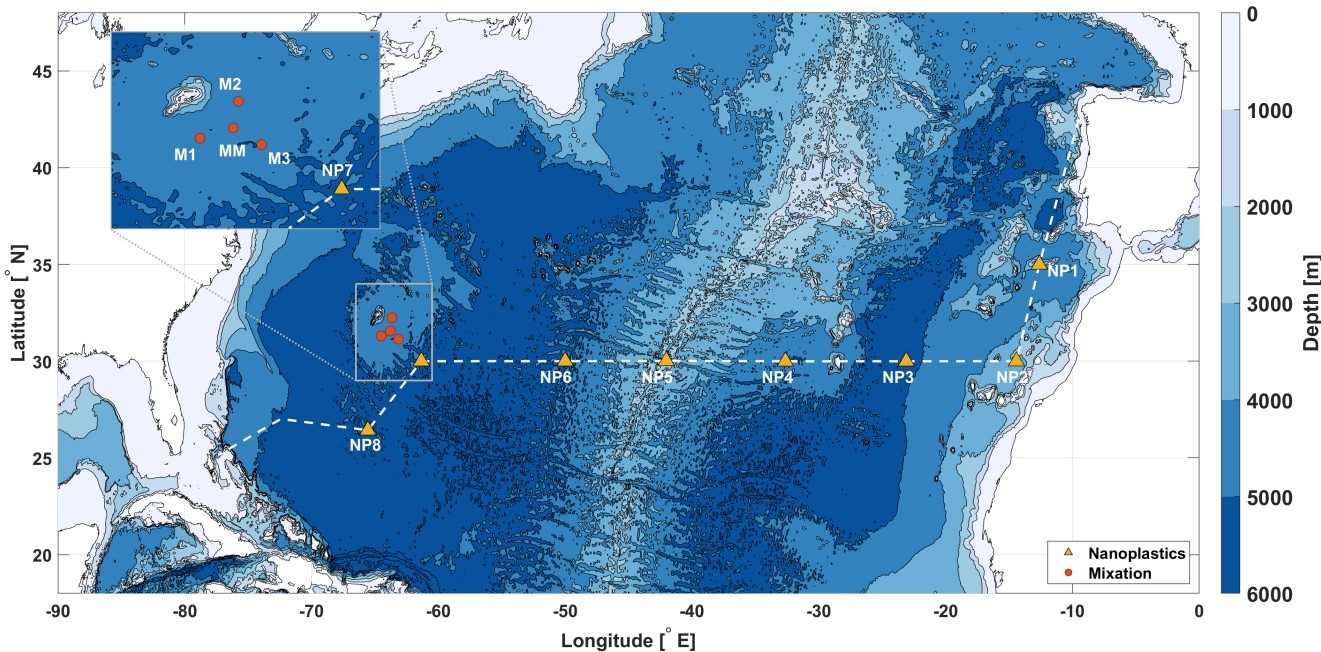

**Figure 1.** Map with the locations of the stations of the Mixation 2 project (*R/V Atlantic Explorer*, Aug 2023, red circles) southeast of Bermuda, and the stations of the Nanoplastics 2 project (*R/V Pelagia*, Nov-Dec 2023, orange triangles), roughly along the transect along $30°$ N.

of one of the probes. These problems were taken care of in the post-processing and at any time, at least one of the probes was functioning as intended. The stations of the Mixation cruise are referred to as M1,M2,M3 and MM. The stations of the Nanoplastics cruise will be called NP1 - NP8. An extensive description of all used methods is provided in the appendices.

All (in-situ) temperature and conductivity data from the CTD systems were converted to Conservative Temperature ($\Theta$) and Absolute Salinity ($S_A$) using the GSW software toolbox (IOC et al., 2010; McDougall and Barker, 2011) and Neutral Density $\gamma^n$ Jackett and McDougall (1997). Conservative Temperature $\Theta$ ($°C$) is proportional to potential enthalpy (by the constant heat capacity factor $c_p^0$ ($J\ kg^{-1}\ K^{-1}$), representing the heat content per unit mass of seawater (McDougall, 2003; Graham and McDougall, 2013). Absolute Salinity $S_A$ is designed to approximate the ratio between the mass of dissolved material and the mass of seawater ($g\ kg^{-1}$) (Wright et al., 2011; McDougall et al., 2012). It is measured on the reference-composition salinity scale (Millero et al., 2008).

In this manuscript various methods to estimate the dissipation rate are being used. To avoid confusion in the comparison of the estimates for $\varepsilon$ we will use the following subscripts: $_{\mu U}$ will be used to indicate variables linked to the microstructure *shear* data, similarly, $_{\mu T}$ will be used for the data belonging to the microstructure *thermistor* data. The subscript $_{FS}$ will be used for the finescale strain method and $_{TP}$ will be used for the Thorpe resorting.

## 3 Energy dissipation rate from microstructure measurements

The microstructure measurements made at each station (Fig. 2) consist of observations made by two types of sensors, the airfoil shear-probes and the fast-response thermistors. Both type of sensors measured at a rate of 512 Hz and provide a way to obtain the turbulent dissipation rate of kinetic energy $\varepsilon$ and thermal variance $\chi_\Theta$. $\varepsilon$ values were obtained from shear as well as temperature measurements. $\varepsilon_{\mu U}$ and $\chi_\Theta$ are obtained from spectral integration of the microstructure velocity and temperature data (Kraichnan, 1968; Nasmyth, 1970; Oakey, 1982). $\varepsilon_{\mu T}$ follows from fitting the temperature gradient spectrum to a theoretical Kraichnan spectrum and the fitted Batchelor wavenumber $k_B$ (Kraichnan, 1968; Ruddick et al., 2000). A full description of the processing of the microstructure data and the applied quality control checks is provided in Appendix A. We present results from both methods and compare them, showing that for a large part of the deep ocean, shear-based estimates may overestimate dissipation rates because the actual dissipation is lower than their noise floor. Such observation motivates us to investigate the range of turbulence levels that can be resolved by each type of sensors, and discuss how shear and temperature-based $\varepsilon$ estimates may be combined to obtain a single profile of energy dissipation rates.

### 3.1 Determining the noise floor for shear-probes

For most of the ocean, e.g. more energetic environments and the upper ocean, there is in general a good agreement between energy dissipation rates measured by the shear probes ($\varepsilon_{\mu U}$) and thermistors ($\varepsilon_{\mu T}$), usually within a factor of 2 (Oakey, 1982; Scheifele et al., 2018; Piccolroaz et al., 2021; Yasuda et al., 2021). In less energetic environments, e.g. the abyssal ocean over smooth topography, there is often a larger discrepancy between the two types of probes (Fig. 2). These discrepancies are particularly noticeable in the deep parts of station NP3, below 2000 m, where $\varepsilon_{\mu U}$ saturates at $\sim 10^{-10}$ W kg$^{-1}$ whilst $\varepsilon_T$ reaches values as low as $\sim 10^{-12}$ W kg$^{-1}$.

Such differences are clearly reflected in the $\varepsilon$ probability distributions. While the dissipation rates from the thermistors have the expected nearly-symmetric and approximately log-normal distribution found for ocean turbulence (Cael and Mashayek, 2021; Gregg et al., 1993), the distribution of the shear dissipation rates is more heavily skewed (Fig. 3a,c). The skewness of the distribution of $\varepsilon_{\mu U}$ is here attributed to the noise floor of the shear probes. In a low-energetic environment as this, the presence of the noise floor can bias the estimated dissipation towards larger values, by removing the left tail of the distribution. Often, the noise floor for shear measurements is reported to be $\mathcal{O}(10^{-10})$ W kg$^{-1}$ (e.g., Fer et al., 2014) and is attributed to contamination of the shear spectrum by electronic noise of the instrument. The exact magnitude of the noise floor varies not only between instruments, platforms and manufacturers, but also between cruises due to for example small differences in instrument set-up (which is why for Fig. 3 the distributions are plotted separately for each cruise).

We now explore various methods for defining the noise floor of shear-derived $\varepsilon_{\mu U}$ for each dataset. Scheifele et al. (2018) suggests using the mode of $\varepsilon_{\mu U}$ as an indicator for the noise floor, whereas Piccolroaz et al. (2021) suggests taking the lower $5^{th}$ percentile. The noise floors derived from these methods for both cruise datasets are summarized in Table 1 and marked in Fig. 3b,d. However, because of the large difference between the estimates of dissipation rates from the fast-response thermistor and the shear probes for station NP3 (below 2000 m, Fig. 2), we have a unique opportunity to provide another estimate of the noise

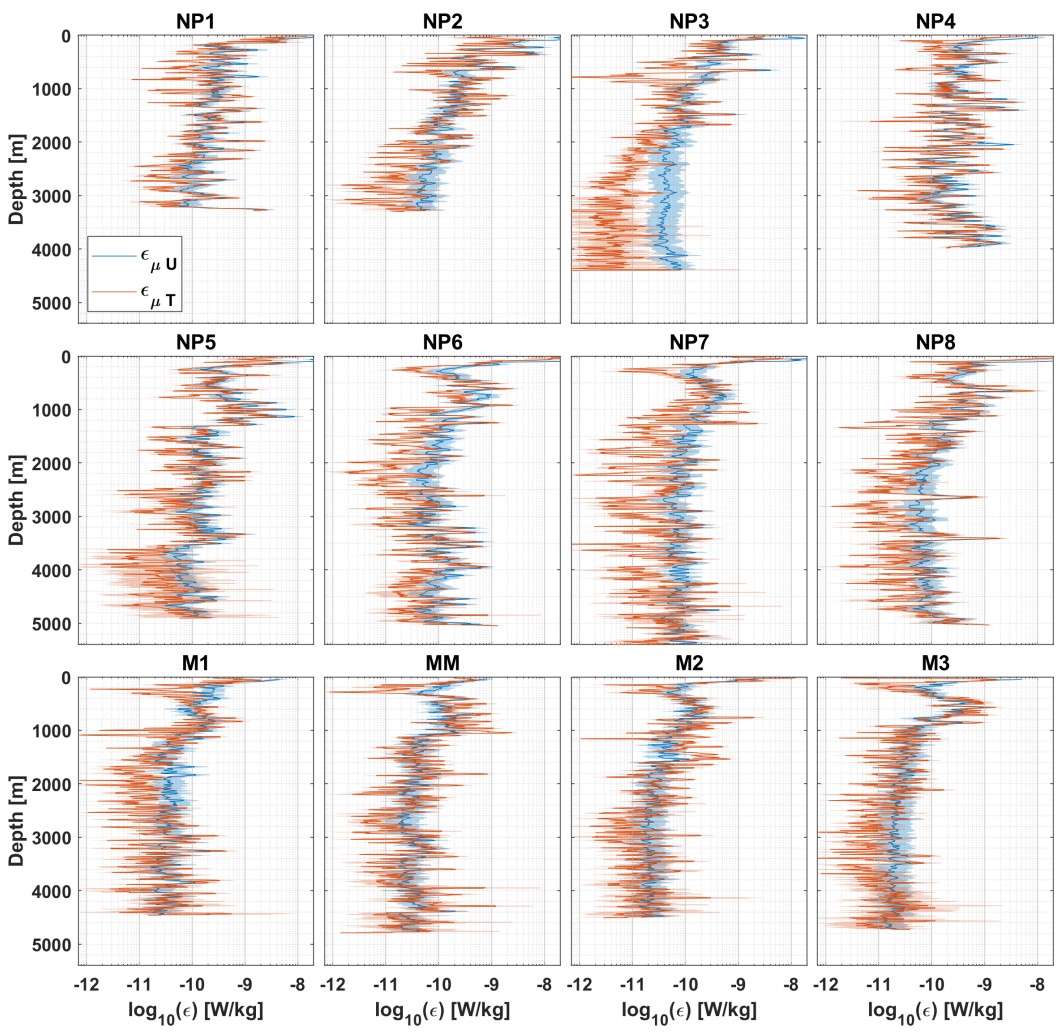

**Figure 2.** Dissipation rates obtained from the microstructure shear probes (in blue) and the microstructure thermistor probes (in red) for each station. The background shading marks the 95% confidence interval. A 5-point running mean has been applied to increase readability.

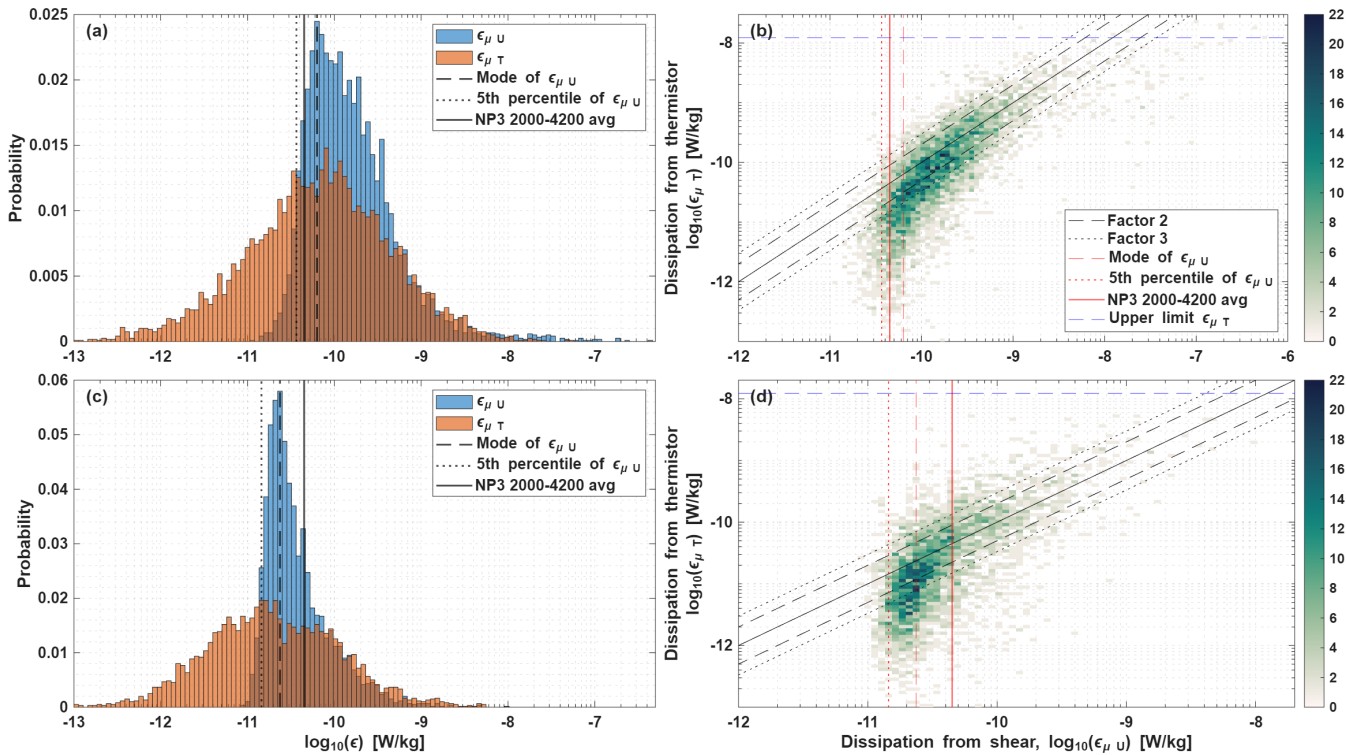

**Figure 3.** (a) Distributions of the $\varepsilon_{\mu U}$ and $\varepsilon_{\mu T}$ data from the Nanoplastics II stations. (b) Cross-validation between $\varepsilon_{\mu U}$ and $\varepsilon_{\mu T}$ for the Nanoplastics II data. (c,d) are the same as (a,b) but for the Mixation II data. Panels (b,d) the solid black line notes the 1:1 ratio. Other black lines mark a factor 2 and 3 difference between the sensors. Shear probe noise floors according to Table 1 are marked in red. A clear break in the agreement can be seen around the level of the noise floor of $\varepsilon_{\mu U}$.

floor for shear probes. The open-ocean turbulent dissipation rates are so low in this region, that the estimates of dissipation by the shear probe are about an order of magnitude larger than the dissipation rates measured by the thermistors, and are therefore reflective of the noise in the instrument. We can thus estimate the noise floor empirically by taking the average and standard deviation over part of such noise-dominated region, where the disagreement between the $\varepsilon_{\mu U}$ and $\varepsilon_{\mu T}$ is largest and consistent (i.e. st. NP3 between 2000-4200 dbar). The (arithmetic) mean over this section is $4.5 \times 10^{-11} \pm 1.9 \times 10^{-11}$ W kg$^{-1}$.

Table 1 shows that the empirical estimate is close to the methods used by Scheifele et al. (2018) and Piccolroaz et al. (2021), ranging $1.4 - 6.3 \times 10^{-11}$ W kg$^{-1}$. It is noted that the Nanoplastics stations have higher noise levels than the Mixation dataset, even though the same instrument was used and that this empirical noise estimate is rather conservative for the Mixation cruise, given the solid red line in Fig. 3d. However, the noise floor of a sensor is not a hard threshold, it represents the range where the signal is dominated by an increasing amount of noise. Instead, to be sure the signal contains very little noise, we

suggest to set the threshold for discarding shear-based dissipation rates at the mean plus two standard deviations, which equals $8.3 \times 10^{-11}$ W kg$^{-1}$. When also accounting for other environmental and instrument-specific factors influencing uncertainties

**Table 1.** Noise floor of $\varepsilon_{\mu U}$ by method and split between datasets. Values are expressed in $10^{-11}$ W/kg.

| Method | Mixation II | Nanoplastics II |
|---|---|---|
| Scheifele et al. (2018) | 2.4 | 6.3 |
| Piccolroaz et al. (2021) | 1.4 | 3.7 |
| Empirical | | $4.5 \pm 1.9$ |

mentioned above, a value of $1 \times 10^{-10}$ W kg$^{-1}$, as often found in literature, would equal the mean plus 3 standard deviations and would put a user absolutely at the save side of knowing the results are not dominated by noise.

### 3.2 Determining limits for the fast-response thermistors

The thermal variance dissipation rate and the Batchelor wave number were obtained by fitting the observed microscale temperature-gradient spectrum to a theoretical model spectrum (Kraichnan, 1968; Ruddick et al., 2000) (see App. A for a more extensive description).

Then, $\varepsilon_{\mu T}$ is derived from the fitted Batchelor wave number ($\varepsilon_{\mu T} = \nu \kappa_T^2 k_B^4$), which signals the position of the spectral roll-off at high wave-numbers (Ruddick et al., 2000). Examination of the $\varepsilon$ profiles and probability density function suggest that

the noise level of the thermistor, which is known to be much lower than that of the shear probe (Kocsis et al., 1999; Scheifele et al., 2018), has not significantly impacted our observations. In our processing routines we discard any data segment where the average spectral levels fall below a factor of 1.3 (Piccolroaz et al., 2021) of the well-characterised sensor noise curve (following RSI, see Technical Note 40 available at www.rocklandscientific.com). Only 0.49% of data-segments were discarded following that criterion. For the retained data, the distributions of $\varepsilon_{\mu T}$ do not show a strong deviation from the expected log-normal

distribution (Fig. 3a,c), as found for the shear sensor, suggesting that these estimates are still reliable and above the noise floor.

Of bigger concern for the thermistor-based estimates are an upper limit due to the limited time-response of the thermistor (5-10 ms), which smoothest out high-frequency temperature fluctuations. Whilst the limited, speed-dependent time-response is corrected for using a double-pole transfer function (Vachon and Lueck, 1984), a harder high-wavenumber cut-off is imposed by the fact that an anti-aliasing filter is applied to the microstructure temperature data. The filter is applied at $\frac{1}{5}^{th}$ of the sampling

frequency ($512\,Hz$), at $\sim 100$ Hz The filter causes a rapid roll-off of the temperature gradient spectrum, similar to the physical roll-off occurring around the Batchelor wave number $k_B$. The anti-aliasing will cause the observed spectra to display an unphysical roll-off at a wavenumber of $k_A = f_A/W = (0.2 \times 512)/0.8 = 128\,cpm$, where $W = 0.8$ m/s is the profiling speed. When $k_B$ is smaller than $k_A$, the spectrum will have two roll-offs, of which one is physical at the Batchelor wave number, and the second is instrumental at $k_A \approx 128\,cpm$. In this case, the fitting algorithm will capture the physical roll-off as it should.

For the case when $k_B > k_A$, the fitting algorithm will fit to the instrumental roll-off, rather than the physical roll-off and the resulting $k_B$ estimate will underestimate the actual $k_B$. Assuming that the highest $k_B$ that can be correctly used is $k_B \approx k_A$, this would result in an upper limit for the thermistor estimates at $\varepsilon_{\mu T,max} = \nu \times \kappa_T^2 \times (2\pi k_B)^4 = \mathcal{O}(1 \times 10^{-8})$ W kg$^{-1}$). With $\kappa_T$ being the thermal molecular diffusivity.

### 3.3 Combining shear-based and thermistor-based dissipation rates

The discussion above shows that the shear probes perform well above the noise floor of $8.3 \times 10^{-11}$ W kg$^{-1}$, whereas the thermistor probes provide good estimates of the dissipation rate below $1 \times 10^{-8}$ W kg$^{-1}$ (Fig. 3). The temperature-based dissipation rates $\varepsilon_{\mu T}$ show that for large parts of the open ocean, the actual dissipation rate can be up to two order magnitude smaller than the noise floor of shear probes. This highlights that awareness of the sensor limitations is important for using the right sensor to get reliable results. Something that is often overlooked and suggests that shear probes and thermistor probes

should be combined to get the optimal result in regions of weak turbulence.

     For combining both instruments, we note that the shear probes provide the most direct method for obtaining $\varepsilon$, while estimating the dissipation rate from the thermistor probes requires more assumptions and is more indirect. Hence, when possible, the shear-based estimates are preferred over thermistor-based estimates. We will therefore use shear-probe estimates above the empirically determined threshold of $8.3 \times 10^{-11}$ W kg$^{-1}$. If estimates are below this, thermistor-based dissipation rates are

used instead. This results in a combined profile of dissipation rates, referred to as $\varepsilon_{\mathrm{vmp}}$, for each station. We will use $\varepsilon_{\mathrm{vmp}}$ when comparing the finescale and Thorpe parameterizations with microstructure data in the remainder of the manuscript.

## 4 Finescale parameterization

We here apply the finescale parameterization to estimate $\varepsilon$. The finescale parameterization assumes that the observed variance in the strain or shear spectra at the fine scale $\mathcal{O}(10-100)$ m are predominantly caused by the breaking of internal gravity waves, and that the dissipation rate of turbulent kinetic energy follows from nonlinear wave-wave interactions that transfer energy to smaller scales, that ultimately break into turbulence (Thompson et al., 2007; Whalen et al., 2015; Dematteis et al., 2024). Such estimates are obtained using vertical gradients of properties such as $\Theta$, that can be obtained from standard CTD data. This method can be either applied to shear, strain or shear and strain data (Gregg, 1989; Polzin et al., 1995, 2014). Since large-scale shear data is absent in this dataset, we consider two calculations of the finescale strain-only method, with the strain rate based on the buoyancy frequency $N^2$ and vertical temperature gradients $\Theta_z$. The dissipation rate $\varepsilon_{FS}$ is obtained via (Whalen et al., 2015)

$$\varepsilon_{FS} = \varepsilon_0 \frac{\overline{N^2}}{N_0^2} \frac{\langle \xi_z^2 \rangle^2}{\langle \xi_{zGM}^2 \rangle^2} h(R_\omega) L(f, N). \tag{1}$$

Here $\langle \xi_z^2 \rangle^2$ is the strain variance and $\overline{N^2}$ the segment-avaraged buoyancy frequency, obtained via the method of adiabatic levelling (Bray and Fofonoff, 1981). The strain variance is compared to a theoretical Garrett-Munk (GM) reference spectrum $\langle \xi_{GM}^2 \rangle^2$ (Garrett and Munk, 1975; Cairns and Williams, 1976). Furthermore, $\varepsilon_0 = 6.73 \times 10^{-10}$ m$^2$ s$^{-2}$ and $N_0 = 5.24 \times 10^{-3}$ rad s$^{-1}$ are constants (Whalen et al., 2015), and $L(f, N)$ and $h(R_\omega)$ are correction functions for latitudinal dependences and the shear-to-strain ratio (Polzin et al., 1995; Gregg et al., 2003). The reader is referred to App. B for a full description of these functions.

### 4.1 The shear-to-strain ratio $R_\omega$

In order to get estimates using the finescale strain parameterization, in the absence of shear data, a choice needs to be made for the shear-to-strain ratio $R_\omega$. Different studies have used available shear data to calculate this value and reported a large range of values for $R_\omega$, both in space and time, ranging from 1-50 (Waterman et al., 2013; Chinn et al., 2016; Fine et al., 2021), while Kunze et al. (2006) reports an average oceanic value of $R_\omega = 7$. $R_\omega$ for the GM model spectrum is $R_\omega = 3$, which corresponds to $h(R_\omega) = 1$ (Eq. (B4)). It can be argued that the best choice to gain optimal results would be to use a variable $R_\omega$ (e.g Sun et al. (2024)), as the dissipation rates (and thus the error) are directly proportional to the choice of $R_\omega$. Though in practice, often is chosen for either $R_\omega = 3$ or $R_\omega = 7$. With our dataset a direct comparison with microstructure data can be made and so the better choice for $R_\omega$ can be assessed. To this end, the results for three different choices of $R_\omega$ are compared to the ratio $\varepsilon_{FS}/\varepsilon_{\mathrm{vmp}}$, for both calculations of the strain rate (Fig. 4). Estimates of $\varepsilon_{\mathrm{vmp}}$ have been averaged over the same bins as the finescale method. Arguments can be made for either choice for $R_\omega$ (3 or 7). Though for both methods, $R_\omega = 3$ gives the best fit overall, as indicated by the median of the estimates (middle orange line in the figure). Which for both methods falls within a factor of 2. Lastly, it can be seen that the estimates $\varepsilon_{FS,\Theta}$ tend to overestimate the estimates $\varepsilon_{\mathrm{vmp}}$, whereas the estimates of $\varepsilon_{FS,N^2}$ contain both over- and underestimates.

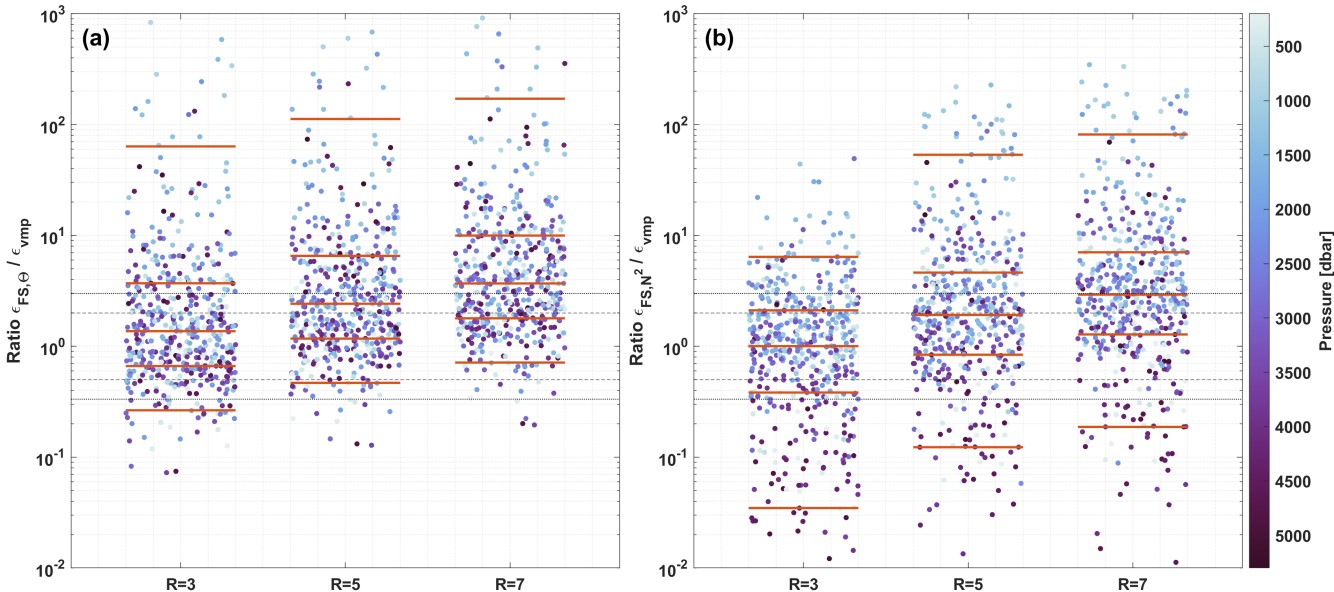

**Figure 4.** The ratios between $\varepsilon_{FS}$ and $\varepsilon_{vmp}$, for different values of $R_\omega$. With (a) strain based on $\Theta_z$ and b) strain based on $N^2$. $\varepsilon_{vmp}$ has been averaged over the same segments as the segments of the finescale method. Color coding of the dots indicates average pressure of finescale estimate. The black lines indicate a factor 2 (dashed) and 3 (dotted) respectively, and the orange lines mark, from bottom to top, the 5,25,50,75 and 95th percentiles. Jitter of the dots is to increase readability.

## 4.2 Dissipation rates from the finescale parameterization

In Fig. 5, the ratio $\frac{\varepsilon_{FS}}{\varepsilon_{vmp}}$ is plotted per station. Here the upper 100 dbar is omitted, as well as the two bins closest to the bottom,
due to boundary effects (note, the surface mixed layer was shallower than 100 dbar for all profiles). Note that station NP5 is located within a fracture zone canyon with an observed bottom boundary layer from 3800 dbar and deeper, such that processes other than internal wave breaking may dominate the dissipation (Kunze et al., 2002; Polzin et al., 2014), and results should be treated with caution.

    In general the two different methods (temperature-strain and density-strain) tend to reproduce similar patterns of overesti-
mating or underestimating $\varepsilon_{vmp}$, though there are some areas with distinctly different patterns. The first area is at the depth of the Mediterranean Outflow Waters (st. 1-3, 800-1500 dbar), where both methods overestimate the dissipation rate, and in particular the temperature-based estimates. Another region is at station M2 between 1000-2000 dbar, where large overestimates are observed for the temperature-based method. The finescale method with strain calculations based on vertical temperature gradients is known to be sensitive for finestructure of water-mass variability (Kunze et al., 2006; Thompson et al., 2007).
Finestructure watermass variability in the form of interleaving patterns are observed in the T/S diagram (Fig. B1). The depths where these patterns are observed correspond to the locations where temperature-based finescale estimates (severely) overes-

timate the measured dissipation rates $\varepsilon_{\mathrm{vmp}}$. This finestructure can be caused by double-diffusive processes (e.g. salt-fingering, Laurent and Schmitt (1999)) or isopycnal stirring of tracers (e.g. Ferrari and Polzin (2005)), instead of internal waves as assumed in the method. This decouples the temperature strain from the density strain ($N^2$) making it unsuitable for estimating dissipation (Ferris et al., 2022).

Another important region of discrepancy are the deep abyssal regions, where especially the finescale method based on the buoyancy frequency underestimates the dissipation rate. For small $\Theta$ and $S_A$ gradients, the strain rates based on $N^2$ become sensitive to salinity spiking (Kunze et al., 2006; Thompson et al., 2007; Whalen et al., 2015). Changes in salinity as measured by the conductivity sensor are then dominated by sensor noise. Temperature sensors have higher accuracy than conductivity sensors and thus do not experience these problems in the regions as seen in (Fig. 5a, see also Sec. 5). Based on these results, we suggest to discard segments for which the maximum salinity difference (over the segment) is smaller than 0.008 g kg$^{-1}$ and maximum temperature difference is smaller than $0.08°C$ (marked by yellow dots in Fig. 5b). Note this is empirically estimated for this dataset, so slightly different values might be needed for other areas.

An alternative explanation for the underestimates can be found in the choice for $R_\omega$. If the actual $R_\omega$ is higher than the chosen value of $R_\omega = 3$, it would cause an underestimate in the dissipation rate. One of the causes for the higher $R_\omega$ can be the interaction of flow with topography and lee wave generation (Ferris et al., 2022). Though most of these profiles are taken over abyssal plains with relatively smooth topography, we consider this factor of limited influence to the underestimates.

Overall, the finescale estimates tend to approximate the dissipation rate reasonably well when compared to $\varepsilon_{\mathrm{vmp}}$, also in this weak turbulent regime. By taking $R_\omega = 3$, we find that 75% of $\varepsilon_{FS,N^2}$ is within a factor 5 of $\varepsilon_{\mathrm{vmp}}$. Similarly, 79% of $\varepsilon_{FS,\Theta}$ is within a factor 5 of $\varepsilon_{\mathrm{vmp}}$. These results illustrated the caveats of the two different methods of calculating the strain rate. Though both methods perform comparably well in terms of accurately estimating the dissipation rate, the finescale method with strain based on the buoyancy frequency is preferred for the upper parts of the watercolumn, but for abyssal segments and areas with weak (salinity) gradients, the finescale method with strain based on temperature gradients appears to perform better.

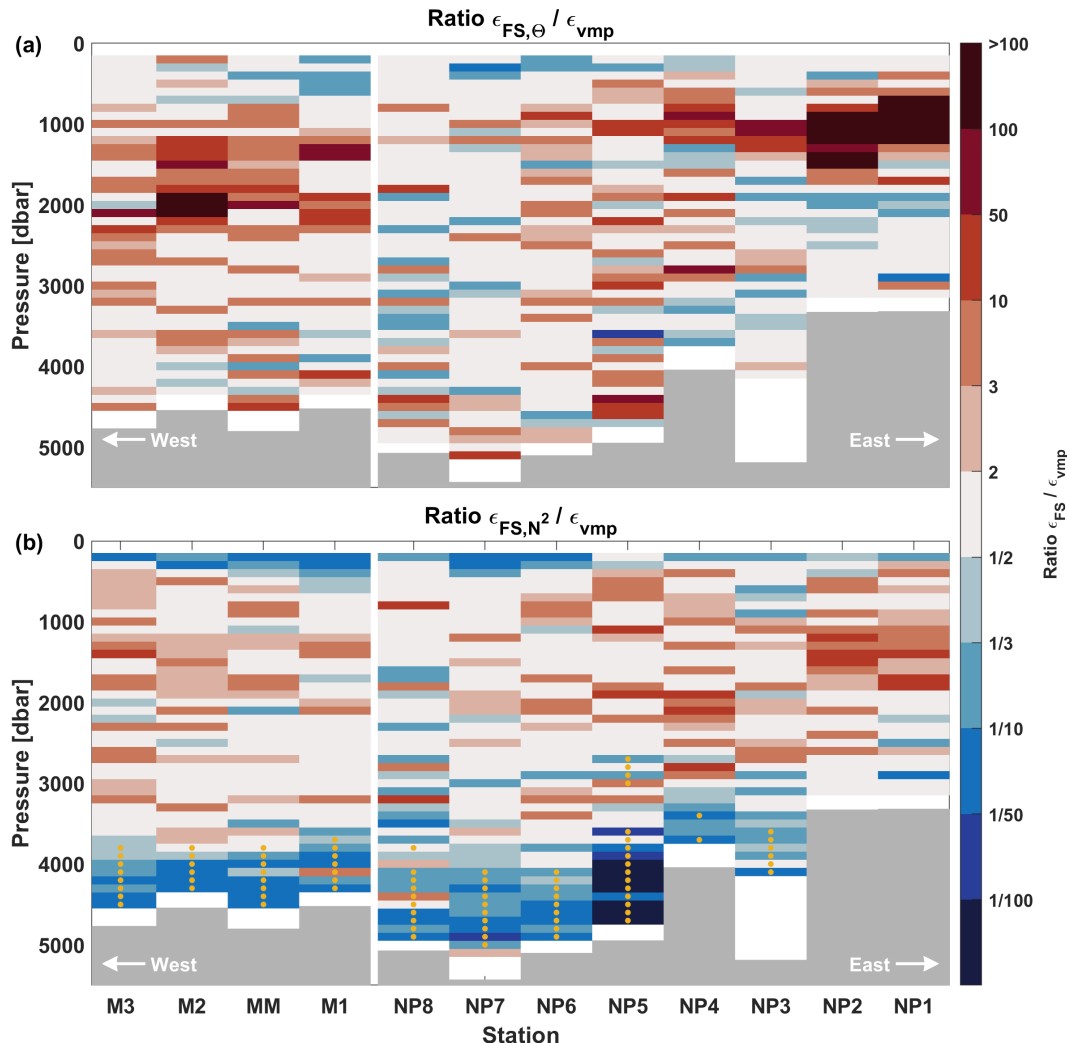

**Figure 5.** Ratios of the finescale method and the microstructure data (averaged over the same bins). The surface mixed layer and the bottom two estimates of the finescale method are omitted due to boundary effects. With for (a) finescale strain based on $\Theta_z$ and (b) finescale strain based on $N^2$. Yellow dots mark the estimates flagged by the threshold stated in the text.

## 5 Thorpe Resorting

The size of vertical overturns of density, where higher density is found above lower density, can be associated with the dissipation of turbulent kinetic energy $\varepsilon$ (Thorpe, 1977; Dillon, 1982). The size of the overturns is characterised by the Thorpe length scale $L_T$, usually defined as the root-mean-square (rms) of the Thorpe displacements: the shortest distance a fluid parcel needs to move in the vertical direction, in order to be at the location where it is stably stratified according to the stable background conditions. The Thorpe scale is assumed to be proportional to the Ozmidov scale $L_O = \sqrt{\varepsilon N^{-3}}$, which represents largest scales of isotropic motions that are possible before stratification becomes a limiting factor (Dougherty, 1961; Ozmidov, 1965), leaving :

$$\varepsilon_{TP} = a^2 \overline{L_T^2 N^3}, \tag{2}$$

where $a = \frac{L_O}{L_T}$. The method is most commonly applied to temperature overturns, because of the higher accuracy of temperature measurements compared to salinity and thus density measurements. This however, requires additional criteria to establish whether the temperature overturns actually represent density overturns. Regardless, the Thorpe resorting method has been used in a range of studies, from moored temperature sensors (van Haren and Gostiaux, 2012; van Haren, 2020; van Haren et al., 2020), (ship-based) CTD profiles (Ferron et al., 1998; Gargett and Garner, 2008; van Haren and Gostiaux, 2014) and numerical modelling (Smyth et al., 2001; Taylor et al., 2019), amongst others.

Here we calculate $\varepsilon_{TP}$ from the high resolution CTD (64Hz) mounted on the microstructure profiler, and compare this to $\varepsilon_{vmp}$ (see Sec. 3). First we will aim to provide a "best" estimate of $\varepsilon_{TP}$, using all available data, including the often unavailable $\varepsilon_{vmp}$, which is used as benchmark. In doing so we also investigate what extra criteria are needed in order to allow for the use of temperature overturns as a proxy for density overturns. Then we provide estimates of $\varepsilon_{TP}$ that exclude the use of measured $\varepsilon_{vmp}$, as this information is often not available. Instead we here examine the use of recently available climatological estimates of $\varepsilon$ (de Lavergne et al., 2019, 2020) to find that this may improve estimates of $\varepsilon_{TP}$. In addition we explore possibilities for different values of the proportionality constant "$a$".

### 5.1 Defining overturns and the Thorpe length scale

The raw temperature data from the CTD is first bin-averaged to the highest resolution (smallest vertical bin-width) for which individual measurements are distinguishable from another. This depends on the sampling frequency, the profiler fall speed and the sensor response times and is 0.08 m for this paper (App. C1). For each sample (measurement) in this profile, its vertical displacement is calculated as the difference between its current location and its location in the stably sorted profile. Overturns follow from the vertical integral of these displacements. An overturn is defined as the region between two zero crossings of the integral (Mater et al., 2015). Subsequently, the Thorpe length scale for each overturn is taken as the rms of all vertical displacements within each overturn (App. C2).

## 5.2 Choosing a value for "a"

$\varepsilon_{\text{TP}}$ is proportional to $a = \frac{L_O}{L_T}$, which has previously been determined to be $a = 0.8$ based on arithmetic averages over many overturns and a wide distribution using temperature sensors (Dillon, 1982). Other studies, using geometric means, mostly found values in the range between 0.8 and unity, but values between 0.4 and 1.3 have also been reported (Itsweire, 1984; Crawford, 1986; Ferron et al., 1998; Smyth et al., 2001; Mater et al., 2015). Here we will use two estimates of $a$ for comparison that are, 1) the $a = 0.8$ found by Dillon (1982), and 2) the geometric mean $a$ over all overturns, calculated for each profile separately. The latter estimate is only possible because of the available estimates of $\varepsilon_{\text{vmp}}$ that allows us to calculate $a$ for each overturn. Over the 12 profiles, $a$ varies between $0.80$ and $1.33$ with a mean of $a = 1.0 \pm 0.2$. Note that using an arithmetic mean of $a$ instead, is sensitive to high outliers and consequently overestimates $\varepsilon_{\text{TP}}$ with respect to $\varepsilon_{\text{vmp}}$. Using these two choices for $a$, combined with different sets of criteria that determine if individual overturns are eligible to be used to calculate $\varepsilon_{\text{TP}}$, we will define 4 separate estimates of $\varepsilon_{\text{TP}}$ per profile, that are mutually compared.

## 5.3 Criteria for selecting overturns

Not all detected overturns are suitable for estimating $\varepsilon_{\text{TP}}$ for reasons discussed below. These reasons will be used to define four criteria for calculating $\varepsilon_{\text{TP}}$. The effect of these criteria on the detected overturns of an example station are shown in Fig. C1. Roughly 30% of the detected overturns is flagged by one or more of the criteria.

### 5.3.1 Criterion 1: Thorpe-Scales Vs Overturn Size

When the size of the Thorpe scale $L_T$ is larger than the vertical size of the overturn $\Delta z_{\text{overturn,l}}$, it implies that the fluid parcel needs to travel further than the size of the overturn itself in order to reach stably stratified waters. We consider this impossible and indicative of noise, spikes or other undefined problems. Therefore we only use overturns for which;

$$L_T < \Delta z_{\text{overturn}}. \tag{3}$$

### 5.3.2 Criterion 2: Measurement Accuracy vs Overturn Length

By dividing the precision of the temperature sensor by the vertical gradient of temperature, a minimal vertical length scale $L_\Theta^{\text{error}}$ is determined for which two observations are significantly different from one another. Overturns need to be larger than this minimum length scale. Details regarding the calculation of this length scale can be found in App. C3. We will only consider overturns for which,

$$L_\Theta^{\text{error}} < \Delta z_{\text{overturn}}. \tag{4}$$

### 5.3.3 Criterion 3: A watermass criterion using the stability ratio

Observed temperature overturns have to be representable for density overturns. Situations can occur where a registered temperature inversion is exactly compensated by a salinity inversion, such that there is no density overturn, in which case the

temperature inversion does not represent turbulence. In previous studies methods were used in which they would examine temperature-salinity relationships (Galbraith and Kelley, 1996; Mater et al., 2015) or the goodness of fit of a linear relationship between temperature and density (Gargett and Garner, 2008; van Haren and Gostiaux, 2014), among other methods. Results sometimes required tedious manual examination of the results or hard-to define thresholds for excluding data. Here we use two related, but different criteria to address this problem with the intention to simplify watermass based selection criteria. First, we take the Pearson correlation coefficient $\mathrm{CC}_{\Theta,\rho_{\mathrm{loc}}}(l)$ between $\Theta$ and locally referenced potential density $\rho_{\mathrm{loc}}$. Assuming that turbulent overturns work against their local background stratification, the correlation coefficient is calculated over a vertical range of twice the overturn size, with a minimum of 10 m. A coefficient larger than 0.8 or smaller than -0.8, is considered a very good correlation (Asuero et al., 2006). A caveat to this criteria is that salinity might be more important in changing density than temperature, while they both work in the same direction, such that a false correlation between temperature and density could be obtained. Hence, in addition we will use that the stability ratio,

$$R_\rho = \frac{\alpha \frac{\partial \Theta}{\partial z}}{\beta \frac{\partial S_\mathrm{A}}{\partial z}}, \tag{5}$$

indicates the influence of vertical temperature and salinity gradients on the vertical density gradient. When $|R_\rho| > 1$, temperature dominates the changes in density. So for the temperature overturns to be a proxy for density overturns, temperature changes should at the very least be dominant over salinity changes. We combine both criteria to only include overturns for which

$$\mathrm{CC}_{\Theta,\rho_{\mathrm{loc}}}(l) < -0.8 \quad \text{and} \quad |R_\rho| > 1 \tag{6}$$

Further details for calculating and applying this criteria per overturn are provided in App. C. Note that a caveat of using temperature as a proxy for density overturns is that, when temperature is not dominant in setting the density changes, some overturns might be unaccounted for.

### 5.3.4 Criterion 4: Thorpe scale Vs Ozmidov Scale

The fundamental assumption for using the Thorpe scale, is that there is a statistical agreement (over a large number of observations) that the Thorpe scale $L_\mathrm{T}$ is approximately equal to the Ozmidov scale $L_\mathrm{O}$ (Dougherty, 1961; Ozmidov, 1965). When both length scales are not comparable in size, we argue that Eq.(2) does not hold and therefore we only use overturns for which:

$$\frac{1}{5}L_\mathrm{O}(l) < L_\mathrm{T}(l) < 5 \times L_\mathrm{O}(l) \tag{7}$$

Although the factor 5 is somewhat arbitrary, it turns out that this criterion is very important for improving dissipation estimates using Thorpe scales. The stricter the criterion, the fewer dissipation estimates there are, but the looser the criterion, tends to overestimate the dissipation rates (see results below). A factor 5 appears to be a reasonable balance between these two effects.

## 5.4 Different ways for estimating dissipation using overturns.

In this study, we compare the following four methods to estimate the dissipation rate $\varepsilon_{\mathrm{TP}}$ from overturns:

- Method 1 - $\varepsilon_{\mathrm{TP}}^{\mathrm{geo}}$: Here $a$ is the profile-specific geometrical mean, and all criteria are used. Hence, we use all available information.

- Method 2 - $\varepsilon_{\mathrm{TP}}^{\mathrm{Dil}}$: Here $a = 0.8$, and all criteria are used.

- Method 3 - $\varepsilon_{\mathrm{TP}}^{\mathrm{Dilx4}}$: Here $a = 0.8$, and criterion 4 is not used. This is because $\varepsilon_{\mathrm{vmp}}$ is often not available, when CTD measurements are available.

- Method 4 - $\varepsilon_{\mathrm{TP}}^{\mathrm{Dil,clim}}$: Here $a = 0.8$, and all criteria are used. However, criterion 4 is based on climatological parameterized estimates of $\varepsilon$.

Comparing these four estimates allows us to evaluate how well the Thorpe resorting method reproduces microstructure-based dissipation rates. This comparison also helps assess whether the Thorpe method can estimate low dissipation rates in the deep sea (i.e., below the shear probe noise floor), and how the results are affected when criterion 4 is omitted or replaced with climatological values. The latter is of interest as these are globally available, whereas $\varepsilon_{\mathrm{vmp}}$ is usually not. Note that we do not compare a version of method 4 where the geometric mean of $a$ is calculated using the climatological value of $\varepsilon$. This is because the estimates for $\varepsilon_{\mathrm{clim}}$ tend to be larger (Fig. 6) than the instantaneous estimate from the VMP, meaning that $L_{\mathrm{O}}$ and thus $a$ and $\varepsilon_{\mathrm{TP}}$ will be overestimated and thus has no additional value. This also has implications for the application of criterion 4 that we did not further refine.

## 5.5 Results

To compare values obtained in different ways over different vertical depth ranges, we will average $\varepsilon_{\mathrm{vmp}}$ and all $\varepsilon_{\mathrm{TP}}(l)$ estimates (per overturn), into vertical bins of 25 m. This implicitly assumes that the mean $\varepsilon_{\mathrm{TP}}(l)$ over the available overturns within this depth range, is representative of the total dissipation in that depth range. We then assess the Thorpe method by taking the geometric mean and standard deviation of the ratio $\varepsilon_{\mathrm{TP}}/\varepsilon_{\mathrm{vmp}}$, to indicate how well the estimates match. However, due to the nature of the geometric mean, it cannot directly be compared with the factor 2 uncertainty that is known for microstructure measurements. Also, outliers on either end can partially cancel out and are downweighted, so the results of the geometric mean may underestimate the severity of outliers. In addition to the standard deviation, we therefore add another metric that assesses the percentage of $\varepsilon_{\mathrm{TP}}$ that is within a factor 2, 5 or 10 from $\varepsilon_{\mathrm{vmp}}$. This is more indicative of outliers. The results of these metrics are listed in Table 2, for all four methods. An additional cross-validation of the estimates against $\varepsilon_{\mathrm{vmp}}$ is shown in Fig. C2.

**Table 2.** Overview of the performance of methods 1-4 for all data. The 'average ratio' is based on the geometric mean of the ratio $\varepsilon_{\mathrm{TP}}/\varepsilon_{\mathrm{vmp}}$, $\pm$ the geometric standard deviation. The last three columns indicate the percentage of estimates that falls within the respective factor compared to $\varepsilon_{\mathrm{vmp}}$.

| Method | Average ratio | <Factor 2 (%) | <Factor 5 (%) | <Factor 10 (%) |
|---|---|---|---|---|
| 1 $\left(\varepsilon_{\mathrm{TP}}^{\mathrm{geo}}\right)$ | $1.1 \pm 0.8$ | 48% | 87% | 97% |
| 2 $\left(\varepsilon_{\mathrm{TP}}^{\mathrm{Dil}}\right)$ | $1.6 \pm 0.8$ | 54% | 92% | 97% |
| 3 $\left(\varepsilon_{\mathrm{TP}}^{\mathrm{Dilx4}}\right)$ | $2.1 \pm 3.7$ | 38% | 71% | 82% |
| 4 $\left(\varepsilon_{\mathrm{TP}}^{\mathrm{clim}}\right)$ | $1.4 \pm 1.2$ | 45% | 81% | 93% |

### 5.5.1 Comparing Method 1 (Geometric mean $a$) with Method 2 ($a = 0.8$)

Method 1 is closer to $\varepsilon_{\mathrm{vmp}}$ (factor 1.1) than Method 2 (factor 1.6). However, Method 1 has less estimates within a factor 2 (48% vs 54%, respectively) or a factor 5 (87% vs 92%, respectively) than Method 2. This indicates that the Method 2 estimates have a narrower spread, but based on the ratio, there is a systematic overestimation as well. Overestimation of the microstructre estimates can be caused by a too coarse sampling resolution of the CTD instrumentation (Sheehan et al., 2023).

The $a$ factor is a quantity that varies for every overturn. Taking the geometric average over all available data or taking $a = 0.8$ results in dissipation estimates that perform comparable well, as these two $a$ factors are not that different. We have not been able to further identify factors that have a strong influence in determining the magnitude of $a$, as to make a good estimate what $a$ to use.

### 5.5.2 Comparing results when omitting Criterion 4

Both methods 1 & 2 use a direct calculation of $L_{\mathrm{O}}$ in criterion 4 which requires a-priori knowledge of $\varepsilon$. Fully omitting criterion 4, and taking $a = 0.8$, is done by method 3. The impact of these choices is clearly visible in Table 2. Where the average ratio and standard deviation is much larger for method 3 compared to the other methods. Method 3 also has 6 times more outliers beyond a factor 10, than Method 1 or 2, and we assessed that these can sometimes be hundreds of times larger, i.e. large spikes. In short, within the setup of this study, we conclude that criterion 4 that assures that the Ozmidov scale and Thorpe scale are of similar order of magnitude, is of particular importance for obtaining Thorpe-based dissipation estimates that are comparable to direct observations.

Method 4 uses climatological dissipation rates $\varepsilon_{\mathrm{clim}}$ de Lavergne et al. (2019, 2020) for calculating $L_{\mathrm{O}}$ and Criterion 4. Generally $\varepsilon_{\mathrm{clim}}$ slightly overestimates the measured dissipation rates $\varepsilon_{\mathrm{vmp}}$ (Figs. 6 and C2), but overall follows the broad structure of $\varepsilon_{\mathrm{vmp}}$ over multiple orders of magnitude. The climatological estimates do in some occasions show larger differences, e.g. differences in depth (e.g. St. NP5 and MM) or have a larger overestimate (e.g. St. NP2 and NP3).

Method 4 significantly improves the dissipation rates compared to Method 3, where criterion 4 is omitted. The estimates of Method 4 are less accurate and have a larger spread than the estimates of Methods 1 and 2 (Tab. 2), however, the given results seem acceptable. Especially considering that within these errorbars, the results tend to follow the large scale patterns of $\varepsilon_{\mathrm{vmp}}$

over multiple orders of magnitude (Figs. 6 and C2). Using a climatological estimate of the dissipation rate thus provides a way to improve dissipation estimates using the Thorpe resorting method when other dissipation rates are not available.

### 5.5.3 Summary from Thorpe estimates

In this study, we compared dissipation rates estimated from temperature overturns $\varepsilon_{\mathrm{TP}}$ to $\varepsilon_{\mathrm{vmp}}$. With 8 cm vertical resolution of temperature we found $\mathcal{O}(10^3)$ overturns for each profile of which about 30-40% is flagged and removed per profile (mostly by criterion 3 and 4 if applied), based on criteria determining the eligibility of using the overturn for calculating the dissipation rate. For the remaining overturns, dissipation rates $\varepsilon_{\mathrm{TP}}$ are calculated, averaged in 25 m bins and compared against $\varepsilon_{\mathrm{vmp}}$ (Figs. 6 ; Note that while the comparison between methods is done on the 25m-binned data, the figure shows the estimates in half overlapping 200m bins for increased readability. The 25 m binned estimates for three example stations are shown in Fig. C2). Using all available information, roughly $50\%$ of the estimates $\varepsilon_{\mathrm{TP}}$ fall within a factor 2 of $\varepsilon_{\mathrm{vmp}}$, and follow the structure of $\varepsilon_{\mathrm{vmp}}$ quite accurately over multiple orders of magnitude. This indicates the potential for using Thorpe overturns for estimating dissipation rates. However, these results relies heavily on the use of criterion 4, which omits overturns for which the Ozmidov scale is more than a factor 5 different from the Thorpe scale. Without applying this criterion, the results are prone to large outliers and it is difficult, to distinguish between outliers and accurate estimates. However, criterion 4 requires a-priori knowledge of $\varepsilon$ (here obtained using $\varepsilon_{\mathrm{vmp}}$) that usually is not available. When instead using globally available climatological estimates of dissipation, reasonable results are still obtained $81\%$ of the estimates being within a factor 5 of $\varepsilon_{\mathrm{vmp}}$. We also found that for this particular setup, the performance of both methods 1 & 2 was not much different. However, for a different vertical bin-size (now 8 cm) and with other criteria, this may be different.

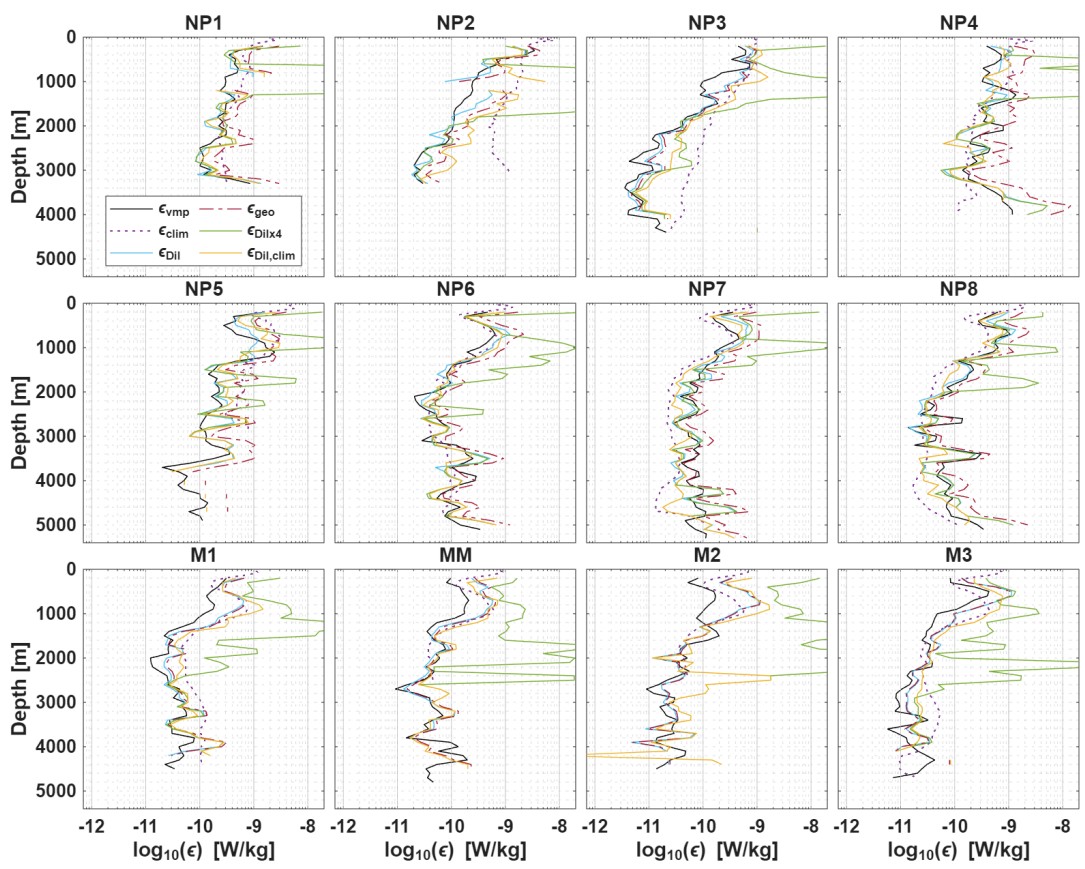

**Figure 6.** Dissipation estimates $\varepsilon_{\text{TP}}$ for all different methods ($\varepsilon_{\text{geo}}$: Method 1, $\varepsilon_{\text{Dil}}$: Method 2, $\varepsilon_{\text{Dilx4}}$: Method 3 and $\varepsilon_{\text{Dil,clim}}$: Method 4, as defined in Sec. 5.4) are compared to $\varepsilon_{\text{vmp}}$ and the used climatological estimates $\varepsilon_{\text{clim}}$. For plotting purposes all data is averaged over the same half overlapping 200m bins.

## 6 The Triple Decomposition Framework

The data from the microstructure profiler provides direct observations of dissipation rate ($\varepsilon$) and temperature variance ($\chi_\Theta$). Using the analytical triple decomposition framework (Stern, 1967; Joyce, 1977; Davis, 1994; Garrett, 2001; Ferrari and Polzin, 2005), the microstructure data can be used to qualitatively assess the relative contributions of isoneutral and dianeutral mixing to tracer variance dissipation (Ferrari and Polzin, 2005; Naveira Garabato et al., 2016; Merrifield et al., 2016; Orúe-Echevarría et al., 2023; Castro et al., 2024). This can shed a light on the dominant processes that set the observed $\Theta$ and $S_A$ variability. A complete description of the framework can be found in Appendix D.

Where a standard Reynolds decomposition separates variables into a mean and a fluctuating component (for any tracer $C$, $C = \overline{C} + C'$), the triple decomposition framework variables are separated into a mean or background component ($C^m$), an mesoscale eddy component ($C^e$) and a microscale turbulent component ($C^t$). One can then write $C = C^m + C^e + C^t$ (Joyce, 1977; Davis, 1994; Garrett, 2001). This separation of scales assumes a spectral gap between the different scales, though it is not clear whether such gap really exists in the ocean (Davis, 1994; Van Haren et al., 1994). A spectral gap is required for the spatially divergent components of the Reynolds stresses. Here we used a simplified approach, ignoring these terms under the assumption of non-divergence amongst others (App. D). This assumption is a source of uncertainty for the method. Nevertheless, the triple decomposition is way to qualitively assess the scale transformations of T-S variance between scales (Ferrari and Polzin, 2005).

The triple decomposition framework implies a balance between the production of tracer variance at the different scales and the dissipation of tracer variance at the microscale. This creates two pathways how tracer variances cascades from the scales where it is produced down to the scales where it is dissipated (Garrett, 2001):

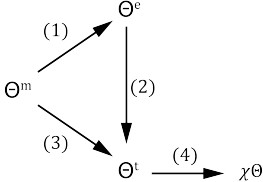

$$\tag{8}$$

In (1), tracer variance is produced by mesoscale stirring. This creates finescale structures that are transferred to the microscale by isotropic turbulence (2) after which it is dissipated (4). In (3), microscale turbulence acts directly on background tracer gradients and variance gets directly transferred from the large scales to the dissipation scales (4). The tracer dissipation itself (end of path) is directly observed by the VMP ($\chi_\Theta$). The total (observed) dissipation results from the combination of downscale variance transfer by small-scale and mesoscale turbulence. At the turbulent scales $\Theta^t$, the dianeutral production term $P_{\overline{\Theta^2}}^\perp$ can be approximated by using a gradient-flux approximation (Osborn and Cox, 1972) and be written in terms of a dianeutral diffusivity $D$ (see also App. D),

$$P_{\overline{\Theta^2}}^\perp = 2\left\langle \mathbf{u}'\Theta^t \right\rangle \cdot \nabla_\perp \Theta^m = -2D|\nabla_\perp \Theta^m|^2 \tag{9}$$

The angled brackets indicate an averaging scale that is large compared to the scales within, but smaller than one scale larger in the framework, here $\nabla_\perp$ is the dianeutral gradient operator. If an assessment can be made of the dianeutral production at the microscale, the contribution by mesoscale stirring $\Theta^e$ follows from,

$$P_{\Theta^2}^{\parallel} = \chi_\Theta - P_{\Theta^2}^{\perp} = \chi_\Theta - 2D\left(\nabla_\perp \Theta^m\right)^2. \tag{10}$$

To obtain $P_{\Theta^2}^{\perp}$ using Eq. (9), it is needed to obtain $\Theta^m$ from measured CTD data $(\overline{\Theta})$. Here an overline refers to data as measured by the CTD instrumentation averaged in $\mathcal{O}(1m)$ vertical resolution. Formally, to get $\Theta^m$ one should be averaging over appropriate long term time and length scales (Ferrari and Polzin, 2005). Only few datasets are large enough and with appropriate spacings that such averaging is possible, e.g. those of NATRE (Ferrari and Polzin, 2005) and DIMES (Naveira Garabato et al., 2016). For our single profiles we instead use the approach of Castro et al. (2024) that approximates background gradients $\left(\frac{\partial \Theta^m}{\partial z}\right)$ using polynomal fits against density, to remove density-compensated temperature intrusions produced from eddy stirring (see App D). Dianeutral diffusivities have been calculated following Eq. (12).

Similar as for $\Theta$, the triple decomposition framework can also be applied with $S_A$ as the tracer. Calculating the dianeutral production term for salinity $P_{S^2}^{\perp}$ from the CTD data is done the same way as was done for temperature. However, the used profiler was not equipped with a microstructure salinity sensor, so haline variance $\chi_S$ has not been measured. To have an estimate of $\chi_S$, we follow the approach of Castro et al. (2024) and assume,

$$\chi_S \approx 2D\left(\frac{\partial \overline{S}_A}{\partial z}\right)^2. \tag{11}$$

The relative roles between isoneutral and dianeutral processes are considered based on the ratios between the dianeutral production and variance dissipation ($P_C^{\perp}/\chi_C$, Fig. 7).

The ratios show a high degree of variability for both temperature and salinity. For the majority of the data, diapycnal production dominates over eddy stirring, especially for temperature. Though there is a signal of more prominent eddy stirring for stations NP1-NP3 between 1000-1500 dbar, which coincides with the influence of Mediterranean Outflow Water. Also a stronger stirring signal is noted around 2000 dbar at station M2. In terms of salinity, a signal of isopycnal stirring dominating over diapycnal production around 1000 dbar extends over the entirety of the dataset. For the stations in the Mediterranean Outflow and from the Mixation dataset (st. M1-M3) the dominance of isopycnal stirring extends to larger depths.

The ratios between the diapycnal production and $\chi$ (Fig. 7) show a similar pattern as to where the finescale estimates overestimate the microstructure derived dissipation rates $\varepsilon_{\mathrm{vmp}}$ (Fig. 5), and also where the Thorpe estimates fail to produce estimates (Fig. 6). Those issues arise where isoneutral stirring processes are of similar or greater importance than dianeutral mixing processes (Fig. 7), and explain increased density-compensated T-S variability at the finescale. Alternatively, double diffusive processes can cause a similar effect as isoneutral stirring. Though almost the entire dataset is susceptible to double diffusion ($R_\rho > 1$), no developed thermohaline staircases are observed in the individual $\Theta$ and $S_A$ profiles, but the T/S diagram (Fig. B1) shows increased interleaving patterns for the areas where a smaller ratios $P_C^{\perp}/\chi_C$ are found, thus supporting the idea that isoneutral stirring processes at least dominate over double diffusive processes. Note that the finescale parameterization assumes that the observed strain is predominantly caused by the non-linear wave-wave interactions (Dematteis et al., 2024).

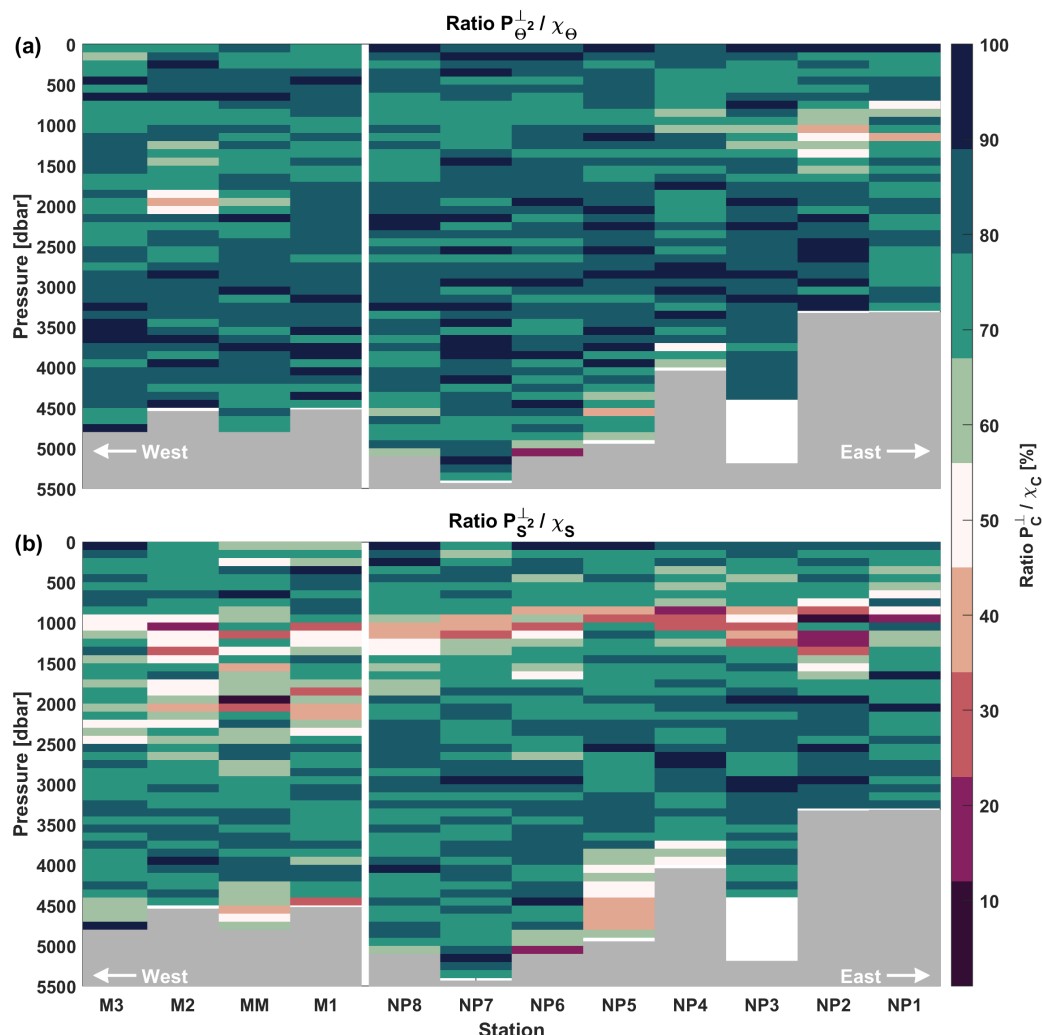

**Figure 7.** The ratios between $P_C^\perp/\chi_C$ expressed in %. With (a) the tracer being temperature and (b) salinity. Ratios have been bin-averaged into 50m bins for readability.

Increased temperature variability will thus cause an overestimate of the strain rate and consequently of the dissipation rate. This effect will be smaller with strain rates based on density variability ($N^2$), as density compensation between T-S variability will
reduce the effects of increased T-S variability due to isopycnal stirring. As such, these locations are also indicative of regions where temperature is not suitable as a proxy for density, as used in the Thorpe method, with perhaps most notably, the Med Outflow around stations NP1-NP3. Although these general patterns of the triple decomposition thus possibly indicate where the Thorpe or finescale method can be applied, a clear metric for rejections of those estimates is yet inconclusive and requires further research.

## 7 Diffusivities

Microstructure data is most commonly used to obtain estimates of the turbulent dissipation rate $\varepsilon$ and of the dianeutral diffusivity $D$. For this we use the Osborn relation (Osborn, 1980),

$$D_\rho = \Gamma \frac{\varepsilon_{\mathrm{vmp}}}{\overline{N}^2}. \tag{12}$$

Here, $D$ is the dianeutral diffusivity ($\mathrm{m^2\,s^{-1}}$), $\overline{N}^2$ is the segment averaged buoyancy frequency obtained by the method of adiabatic levelling (Bray and Fofonoff, 1981). $\Gamma$ is the mixing coefficient and often assumed to be 0.2 (Osborn, 1980; Gregg et al., 2018). Here we calculate $\Gamma$ based on the flux Richardson number $R_f$ (Fernández Castro et al., 2022), using the microstructure data as,

$$R_f = \frac{D_\Theta N^2}{\varepsilon_{\mathrm{vmp}} + D_\Theta N^2}, \quad \text{where} \quad D_\Theta = \frac{\chi_\Theta}{2\left(\frac{\partial \overline{\Theta}}{\partial z}\right)^2}. \tag{13}$$

Here $D_\Theta$ is the temperature diffusivity based on the Osborn-Cox model (Osborn and Cox, 1972), and the effective mixing coefficient is related to the flux Richardson number as (Gregg et al., 2018),

$$\Gamma = \frac{R_f}{1 - R_f}. \tag{14}$$

For the calculation of $\Gamma$, $R_f$ has been smoothed with a 30-point running mean. Most profiles show an increase in $R_f$ between 1000–2000 dbar, leading to a corresponding rise in $\Gamma$, and resulting in differences in diffusivities of up to an order of magnitude (Fig. 8). Areas with increased $R_f$ can be indicative of caveats in the underlying models and assumptions, and can partially be caused by non-turbulence effects such as isopycnal stirring or double diffusion. The resulting diffusivities range from $\mathcal{O}(10^{-6} - 10^{-2})\,\mathrm{m^2\,s^{-1}}$, with increasing diffusivities closer to the bottom (Fig. 8). Enhanced diffusivities extending to mid-depths are observed for the stations at or close to the Mid-Atlantic Ridge (st. NP4-NP6).

Additionally, the triple decomposition framework can also be used to get estimates of the isoneutral diffusivity based on the microstructure data. However, due to the small isoneutral gradients of temperature and salinity in the abyss, the final estimates appear to be noisy and the results inconclusive. The estimates of the isoneutral diffusivity can be found in Appendix D2 for the interested reader.

## 8 Discussion & Conclusions

In this study we analysed a set of full-depth microstructure profiles from the North Atlantic (Fig. 1). First, dissipation estimates from velocity shear probes and the fast-response thermistors were compared (Fig. 2). The thermistor-based estimates generally showed good agreement with shear-based estimates for moderate levels of turbulence, but they were sometimes up to 2 orders magnitude lower than the shear-based dissipation rates for low turbulence levels ($\varepsilon \lesssim 10^{-10}\,\mathrm{W\,kg^{-1}}$). The data suggested that it was unlikely that the noise floor of the thermistor-based estimates was reached, while observations in the weakly turbulent deep ocean were frequently below the noise floor of the shear-based estimates. This could lead to substantial differences when

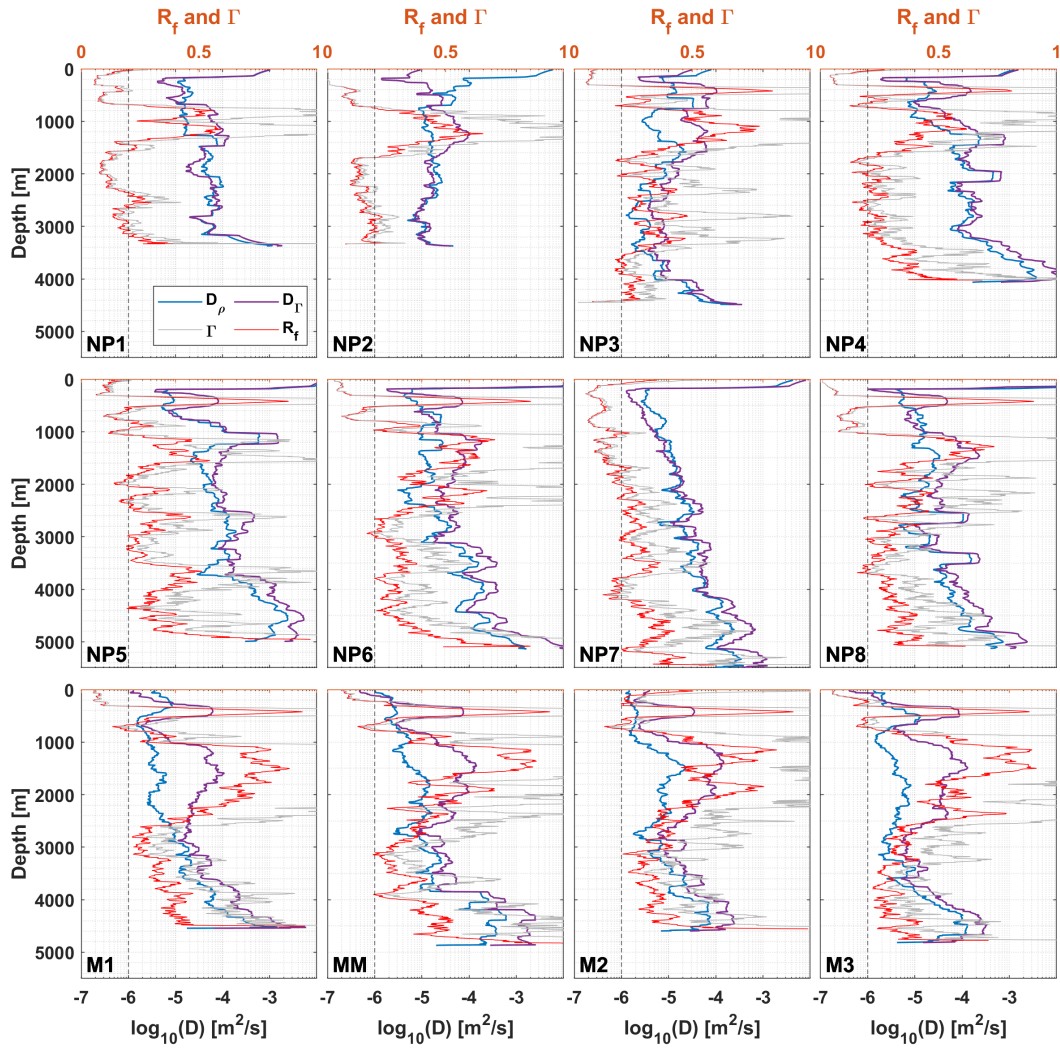

**Figure 8.** The calculated flux Richardson number $R_f$ (in red) and mixing coefficient $\Gamma$ (in grey) for each station. Also shown are the dianeutral diffusivities calculated with an assumed $\Gamma = 0.2$, shown as $D_\rho$. And the diffusivity calculated using Eq. (12), with the variable $\Gamma$, shown as $D_\Gamma$. The canonical $\Gamma = 0.2$ is marked by the dashed black line. A 50-point running average has been applied on all diffusivities for readability.

shear-based dissipation rates are used for e.g. flux calculations, as is common practice. Therefore we combine both products to obtain $\varepsilon$ profiles that adequately capture the range of turbulence levels in our dataset (Fig. 9). Using a coincidental unique profile where dissipation rates from both shear and thermistor based where entirely separated over a large part of the water column (Fig 2, station NP3), we could empirically determine the noise floor for the shear-based method to be at $8.3 \times 10^{-11}$ W kg$^{-1}$. Hence, considering that the upper limit of the thermistor based method is $\sim 1 \times 10^{-8}$ W kg$^{-1}$, we combined both estimates by using thermistor-based dissipation estimates below $8.3 \times 10^{-11}$ W kg$^{-1}$, and shear-based estimates above that. Note that, shear-based estimates are preferred over thermistor-based estimates, when far from the noise floor, as these are arguably more direct and require fewer assumptions. To conclude, when one expects to measure weak turbulence, we recommend using both shear and thermistor probes in measurements to be able to cover the entire turbulent range.

Second, the microstructure $\varepsilon$ estimates were compared to both the finescale parameterization and the Thorpe resorting method, obtained by using the CTD data. The finescale parameterization has been applied in its strain-only form based on strain rates calculated from both the buoyancy frequency and vertical temperature gradients. In general, both methods tend to follow the microstructure estimates reasonably well, over multiple orders of magnitude. However, the temperature-based estimates severely overestimated the measured dissipation rates at the depth of the Mediterranean Outflow Water and stations M2, MM. The triple decomposition framework supported the hypothesis that an increased isopycnal stirring signal was cause for finestructure watermass variability not associated with internal waves, leading to the mismatch. In these locations the method based on the buoyancy frequency performed better. However, the density-based method was affected by salinity spiking for abyssal parts of the profiles due to the small salinity gradients in the segments. Overall we therefore recommend using the temperature-based estimates over the density-based estimates for segments where the maximum salinity difference over a segment is smaller than 0.008 g kg$^{-1}$ and the maximum temperature difference is smaller than $0.08^\circ C$. In general, the agreement in our estimates for $\varepsilon_{FS,N^2}$ compares well with other comparisons between the finescale method and microstructure data (e.g. Whalen et al. (2015); Baumann et al. (2023)). Only few studies have reported on the use of strain based on temperature data (Thompson et al., 2007). In this study, the limitations of this method in upper ocean applications, due to the sensitivity are clearly seen. However, it is also shown that the strength of this method lies in the deeper parts of the watercolumn, where this temperature-based method outperforms the estimates based on buoyancy frequency strain.

For the Thorpe resorting method, the analysis focussed on the selection criteria to distinguish between false and actual overturns. One criterion aimed to provide an objective way for determining if a detected temperature overturn is a good proxy for a density overturns. This criterion is based on the stability ratio $R_\rho$ and a correlation between density and temperature variations (Sec. 5). As the Thorpe method is based on the assumption that the Thorpe lengthscale is proportional to the Ozmidov lengthscale, we tested a second criterion that requires these length scales to be within a factor 5 of each other. The criterion turned out to be critical for removing outliers and providing accurate estimates of the dissipation rate. The caveat here is that this criterion needs a-priori estimates of the dissipation rates for the calculation of the Ozmidov scale. To make this criterion more generally applicable we tested the use of stationary climatologically averaged estimated of dissipation (de Lavergne et al., 2020). It turned out that applying Thorpe based dissipation estimates to CTD data under these criteria, provides very acceptable results (Fig. 9). Various other studies have previously reported on the comparison between the Thorpe method and

microstructure data. Overestimating the dissipation rates by the Thorpe method is often seen and is attributed to either the sampling resolution (e.g. Howatt et al. (2021); Sheehan et al. (2023)) and the relationship between $L_T$ and $L_O$ (e.g. Ferron et al. (1998); Mater et al. (2015)). While sampling resolution depends on the instrumentation used, and can be difficult to improve. With microstructure data, an optimal factor $a$ can be found for the relationship between $L_T$ and $L_O$, but when only coarse data (e.g., CTD data) is available, the use of our criterion 4 and the climatological dissipation estimates is a useful way to improve the Thorpe based estimates, also for areas with weak dissipation rates.

It is noted that the data in this study is relatively sparse. Stations are far apart and only snapshots in time. It is known that for a good comparison between direct microstructure observations and indirect parameterizations (both finescale and Thorpe parameterizations) over larger spatiotemporal scales, the microstructure data should be averaged over equal spatiotemporal scales (Whalen, 2021). The presented results here have first been (vertically) averaged to equal scales, but optimally also horizontally or temporal averaging would be applied if a closer spacing of the stations was available. Adding more profiles will undoubtedly lead to better comparisons between the direct microstructure observations and the indirect parameterizations, but it seems unlikely that it will change the key conclusions of this paper.

In conclusion, both parameterizations approximate the dissipation rates reasonably well, with 79% (for $\varepsilon_{FS,\Theta}$), 75% (for $\varepsilon_{FS,N^2}$) and 81% (for $\varepsilon_{TP}^{clim}$) being within a factor 5 of the measured $\varepsilon_{\text{vmp}}$. Further analyses using the triple decomposition framework (Fig. 7) shows the influence of mesoscale stirring, producing finescale density-compensated TS variance, where also the Thorpe and finescale parameterizations either failed or showed strong overestimates and help understand why the underlying assumptions broke down.

Finally, by using the Osborn relation the dianeutral diffusivities were calculated using $\varepsilon_{\text{vmp}}$. An additional attempt to obtain estimates of the isoneutral diffusivity from combining the dianeutral diffusivities with the triple decomposition framework remained inconclusive.

This work has compared and contrasted several direct observations of dissipation rates with parameterizations (Fig. 9). This helps understand where the use of different parameterizations may or may not be appropriate, and shows the degree to which turbulence can be estimated. Obtaining estimates within a factor 2, even between shear probes, between thermistor probes and between both shear and thermistor probes, is not trivial. In that light, parameterizations that can estimate dissipation rates within a factor 2-5 from measured dissipation rates, could arguably be considered successful. Further improving both observation and parameterizations will allow us to improve global ocean mixing estimates in space and time.

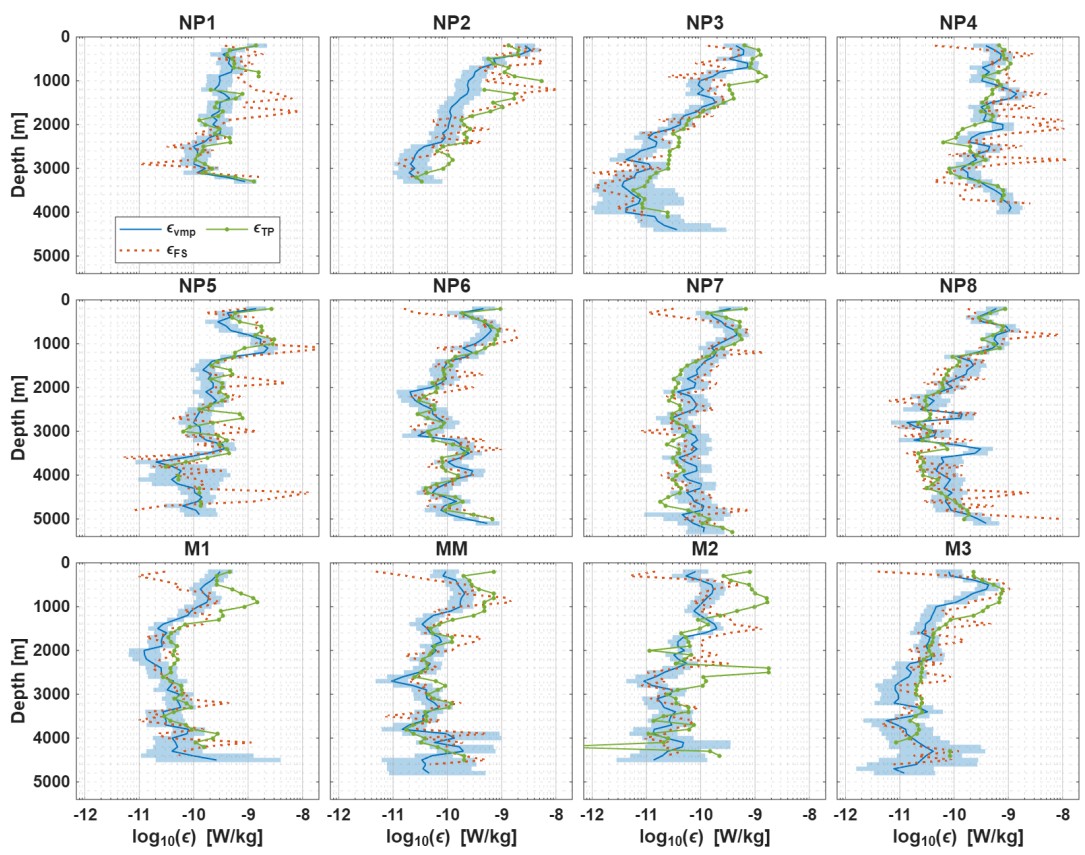

**Figure 9.** The dissipation rates $\varepsilon_{\mathrm{vmp}}$, $\varepsilon_{\mathrm{TP}}$ and $\varepsilon_{\mathrm{FS}}$ plotted for each station. The finescale estimates are combined based on the threshold recommended in Sec. 4. The Thorpe estimates plotted correspond to $\varepsilon_{\mathrm{Dil,clim}}$ of Fig. 6 and method 4 of Sec. 5. All methods are averaged over the same segments as the finescale method.

## Appendix A: Processing of microstructure data

In this appendix, the procedures taken for processing both the microstructure shear and microstructure thermistor data are described.

### A1 Microstructure shear data

The microstructure profiler VMP-6000 of Rockland Scientific International (RSI) was equipped with two orthogonally mounted airfoil shear probes, sampling at a rate of 512 Hz. This provided independent estimates of both $\frac{\partial u}{\partial z}$ and $\frac{\partial v}{\partial z}$. Under the assumption of isotropic turbulence, each component can be independently used for the estimation of $\varepsilon_{\mu U}$ according to (Oakey, 1982; Piccolroaz et al., 2021; Lueck et al., 2024),

$$\varepsilon_{\mu U} = \frac{15}{2}\nu \left\langle \left(\frac{\partial u'}{\partial z}\right)^2 \right\rangle.$$
(A1)

With $\nu$ being the kinematic viscosity and $\left\langle \left(\frac{\partial u'}{\partial z}\right)^2 \right\rangle$ the shear variance, $\frac{\partial u'}{\partial z}$ the horizontal shear fluctuations. (') denotes a turbulent quantity. The processing of the raw microstructure data to $\varepsilon_{\mu U}$ was done using software packages provided by RSI (Zissou Premium, v1.0). This processing involved despiking of the data, highpass filtering and the removal of coherent noise using a Goodman filter (Goodman et al., 2006). Estimates of $\varepsilon_{\mu U}$ were obtained by fitting to a theoretical Nasmyth spectrum (Nasmyth, 1970; Oakey, 1982). Example spectra are shown in Fig. A1a. Furthermore, the estimates were quality controlled

according to the ATOMIX best practices (Lueck et al., 2024). These quality controls check for i) a poor Figure of Merit, ii) a large fraction spikes iii) the ratio between estimates between the two probes, iv) the number of iterations of the de-spiking routine, and v) the amount of variance resolved. Most data was removed by the Figure of Merit criterion, which is a measure of how close the measured spectrum resembles a model spectrum (Lueck et al., 2024). In the limit of very weak turbulence, one or more assumptions underlying the spectra might not be valid, resulting in a poor resemblance and thus rejection of the estimate

based on the Figure of Merit. Most of the rejected estimates originate in regions where the estimates are close to the instrument noise floor. Estimates that did not pass the quality control criteria were removed from further calculations. The estimates from the two different probes were averaged in case both passed the quality control, otherwise the single estimate was taken.

The 95% confidence interval for dissipation estimates obtained by the spectral integration method is given by (Lueck, 2022; Lueck et al., 2024),

$CI_{95\%,U} = \varepsilon_{\mu U}\, exp(\pm 1.96\sigma_{ln\varepsilon}).$
(A2)

$\sigma_{ln\varepsilon}$ is obtained from (Lueck et al., 2024),

$$\sigma_{ln\varepsilon}^2 = \frac{5.5}{1 + \left(\hat{L}_f/4\right)^{7/9}}, \quad \hat{L}_f = \frac{l_\varepsilon}{L_K}V_f^{3/4}$$
(A3)

Where $\sigma_{ln\varepsilon}^2$ is the variance of the dissipation estimate, $L_f$ a non-dimensional datalength, $l_\varepsilon$ the length of the data segment, $L_K$ the Kolmogorov length and $V_f$ the fraction of shear variance that is resolved by the spectral integration. A confidence interval

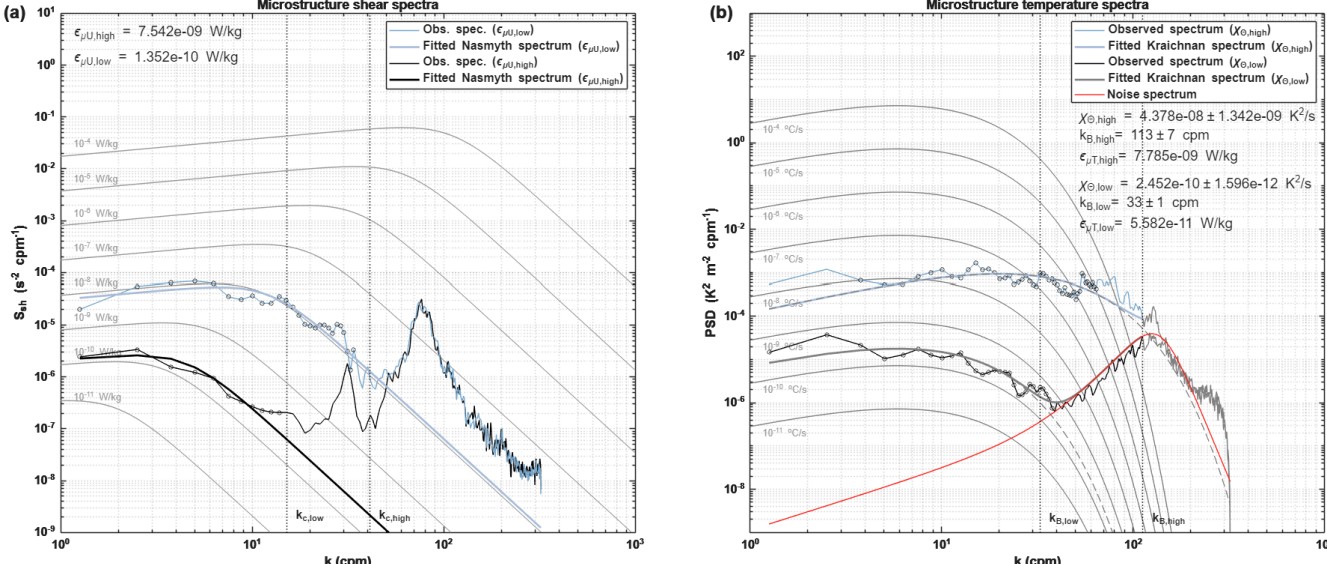

**Figure A1.** (a) Two examples of the observed shear spectra. With the points used in the integration marked by the circles. Background shows the theoretical Nasmyth spectrum for various levels of $\varepsilon$. Vertical dotted line indicates the upper integration limit. (b) Corresponding example microstructure temperature spectra. Points used for the spectral integration are marked by the circles. Background shows the theoretical Kraichnan spectrum for various levels of $\chi$ with the Batchelor wavenumber $k_B$ fixed at 33 cpm. Note, the Kraichnan spectra for $k_B$ fixed at 113 cpm are omitted for clarity. Vertical dotted line shows the Batchelor wavenumber, used for the calculation of $\varepsilon_{\mu T}$.

for dissipation estimates obtained from fitting the inertial subrange follows a slightly different procedure (Lueck et al., 2024), though that does not apply to this dataset, as this generally applies to dissipation rates larger than $\varepsilon_{\mu U} \geq \mathcal{O}(10^{-5} \text{ W/kg})$ (Lueck, 2022).

## A2    Microstructure thermistor data

Simultaneously to the shear probes, the two FP07 fast-response thermistors, sampling at 512 Hz, provide a way to calculate
the dissipation of temperature variance $\chi_\Theta$. Here we will mostly follow the procedure as described by Piccolroaz et al. (2021). Assuming isotropy, $\chi_\Theta$ can be determined from,

$$\chi_\Theta = 6\kappa_T \left\langle \left( \frac{\partial T'}{\partial z} \right)^2 \right\rangle = 6\kappa_T \int\limits_0^\infty \Psi_T(k)dk \tag{A4}$$

Here $\kappa_T$ is the molecular thermal diffusivity and $\left\langle \left( \frac{\partial T'}{\partial z} \right)^2 \right\rangle$ the temperature gradient variance, with $\frac{\partial T'}{\partial z}$ the turbulent temperature gradients. The temperature gradient variance equals the area under the one-dimensional temperature gradient wavenumber
spectrum $\Psi_T(k)$ (Piccolroaz et al., 2021). The measured spectrum is fitted to a theoretical Kraichnan spectrum (Kraichnan, 1968) using the Maximum Likelihood Estimation method of Ruddick et al. (2000). An example spectrum is shown in Fig. A1b.

This method provides an estimate of $\chi_\Theta$, and a way to get $\varepsilon_{\mu T}$ with an uncertainty estimate on the fitted Batchelor wave number $k_B$. Here $\varepsilon_{\mu T}$ depends on the roll-off of the temperature gradient spectrum towards the Batchelor wave number (Ruddick et al., 2000; Luketina and Imberger, 2001). Quality control criteria were applied according to the methods of (Piccolroaz et al., 2021; Ruddick et al., 2000) and were based on:

- the signal-to-noise ratio of the integrated spectrum and the instrument noise spectrum should be larger than 1.3.

- the mean absolute deviation (MAD) over the integrated wavenumber range should be $\text{MAD} \leq 2(2/d)^{1/2}$, with d the number of degrees of freedom of the measured spectrum. The MAD criterion is comparable to the Figure of Merit criterion of the shear data.

- a log likelyhood ratio between a power-law fit and the theoretical spectrum should be $LR > 2$.

Additionally, if estimates from two probes are within a factor of 2.8, they are averaged, otherwise the smallest of the two estimates was taken.

The MLE fitting procedure provides a standard deviation for the fitted Batchelor wave number $k_B$. The dissipation rate $\varepsilon_{\mu T}$ is proportional to the Batchelor lengthscale, which is the inverse of the Batchelor wave number:

$$\varepsilon_{\mu T} = \nu \kappa_T^2 k_B^4. \tag{A5}$$

With $\nu$ the kinematic viscosity and $\kappa_T$ being the thermal molecular diffusivity. Through error propagation, the standard deviation of $\varepsilon_{\mu T}$ follows from,

$$\sigma_{\varepsilon,T} = \left| \frac{\partial \varepsilon_{\mu T}}{\partial k_B} \right| \cdot \sigma_{k_B} = 4\nu \kappa_T^2 k_B^3 \cdot \sigma_{k_B}. \tag{A6}$$

To account for the approximate log-normal distribution of $\varepsilon_{\mu T}$, the variance of those estimates is given by,

$$\sigma_{ln\varepsilon,T}^2 = ln \left( 1 + \frac{\sigma_{\varepsilon,T}^2}{\varepsilon_{\mu T}^2} \right). \tag{A7}$$

Finally, the 95% confidence interval is given by,

$$CI_{95\%,T} = \varepsilon_{\mu T} \, exp(\pm 1.96\sigma_{ln\varepsilon,T}). \tag{A8}$$

## Appendix B:  The finescale parameterization

The finescale strain parameterization is applied using Eq. (1). Here a more extensive description of the formula involved is given.

## B1 Obtaining the observed and model strain variance

As discussed in Section 4, the observed strain rate can be calculated by using either the buoyancy frequency or vertical temperature gradients (Kunze et al., 2006). Here we used both:

$$\xi_{z,N^2} = \frac{N^2 - N_{fit}^2}{N_{fit}^2}, \quad \text{and} \quad \xi_{z,\Theta} = \frac{\Theta_z - \Theta_{z,fit}}{\Theta_{z,fit}}. \tag{B1}$$

Where $\Theta_z$ is the vertical derivative of $\Theta$ over the segment of interest. $N_{fit}^2$ and $\Theta_{z,fit}$ are a quadratic fit to the corresponding variable over the segment. $\overline{N_{fit}^2}$ and $\overline{\Theta_{z,fit}}$ are the mean of the quadratic fits. In this application, we have used vertical 200 meter half overlapping segments, starting at a depth of 100 meter , to avoid the surface mixed layer (the mixed layer is within the 50-80 meter range). Also the two bottom most bins are omitted for boundary effects caused by the bottom boundary layer, where most internal wave breaking occurs.

The strain segments are first detrended and windowed using a Hanning window before being Fourier transformed to obtain the strain spectrum $S_{str}$ of the segment. Each segment is integrated between wavelengths of 10-100 meter to get the strain variance $\langle \xi_z^2 \rangle$. Each segment was integrated between 100 meter wavelength and an upper limit being the highest possible wavelength between 10 and 40 meter, while keeping $\langle \xi_z^2 \rangle \leq 0.2$ as to avoid oversaturation of the spectrum.

The GM model spectrum (Garrett and Munk, 1975), following the GM76 version (Cairns and Williams, 1976), is calculated as (Gregg and Kunze, 1991),

$$\xi_{zGM} = \frac{Eb^3}{2j_*\pi} \left( \frac{N_0}{\overline{N}} \right)^2 \frac{1}{(1 + k_z/k_{z*})^2}. \tag{B2}$$

With,

$$k_{z*} = \frac{\pi j_*}{b} \frac{N}{N_0}. \tag{B3}$$

The used constants are $j_* = 3$, $b = 1300$ and $E = 6.3 \times 10^{-5}$. $\langle \xi_{zGM}^2 \rangle$ is obtained by integrating the GM model spectrum between the same wavelengths as $\langle \xi_z^2 \rangle$.

## B2 Functions for latitudinal dependence and $R_\omega$

The function $h(R_\omega)$ describes the dependency of the ratio between shear and strain and is given by,

$$h(R_\omega) = \frac{1}{6\sqrt{2}} \frac{R_\omega (R_\omega + 1)}{\sqrt{R_\omega - 1}}. \tag{B4}$$

The choice for a value of the shear-to-strain ratio $R_\omega$ in absence of shear data is discussed in Section 4. By choosing $R_\omega = 3$, Eq. (B4) reduces to 1.

The last term of Eq. (1) is a function to account for the latitudinal dependence of internal waves (Gregg et al., 2003). This latitudinal correction is,

$$L(f,N) = \frac{f \operatorname{arccosh}\left(\frac{\overline{N}}{f}\right)}{f_{30} \operatorname{arccosh}\left(\frac{N_0}{f_{30}}\right)}. \tag{B5}$$

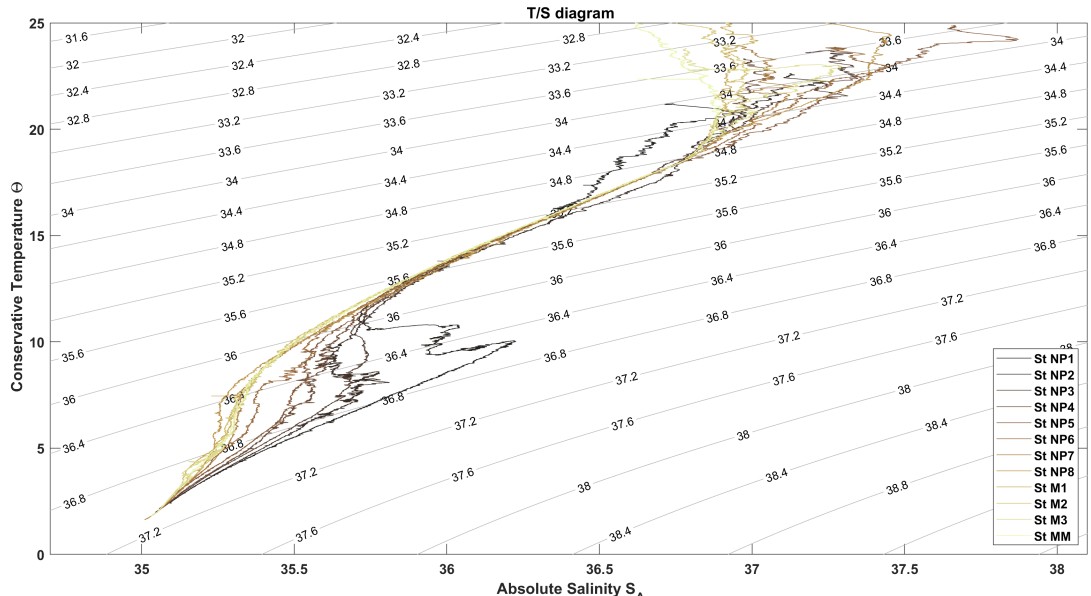

**Figure B1.** A T/S diagram of the different stations. Background contours indicating $\sigma_2$ potential density are provided for reference.

Here $f$ is the Coriolis parameter and $f_{30}$ the Coriolis parameter at a latitude of $30°$.

## B3 Additional figures

A T/S diagram is provided to support the analysis in the main text.

## Appendix C: The Thorpe resorting method

This appendix is concerned with the calculation of temperature overturns and the associated Thorpe lengthscale $L_T$ (Thorpe, 1977) from the CTD sensors mounted on the VMP.

### C1 Defining vertical averaging scale ($\delta z$) based on sensor characteristics

Below we determine a minimum vertical averaging length scale for averaging the measured data from the CTD, which depends on the noise that is dictated by the specifics of the used instrumentation and it interaction with the environment.

### C1.1 Thinnest Overturns

The thinnest overturns that can be detected depend on the sampling frequency of the sensors and the fallspeed of the profiler. The used SBE3 temperature sensor and pressure sensor operated at a frequency of $f_{\mathrm{SBE}} = 64$ Hz. Then, the highest resolution

sampled, $L_z$ (m), is the fallspeed of the profiler $v_{vmp}$ divided by the sampling frequency $f_{\mathrm{SBE}}$. The maximum of $L_z$ over the watercolumn is taken and multiplied by a factor 5 as a sort of "Nyquist frequency" (see Galbraith and Kelley (1996)), such that the thinnest detectable overturn size $L_{thinnest}$ (m) is:

$$L_z = \max \left[ \frac{(v_{\mathrm{vmp}}(z))}{f_{\mathrm{SBE}}} \right] \quad \longrightarrow \quad L_{\mathrm{thinnest}} = 5 L_z. \tag{C1}$$

We find $L_z \approx 0.015m$ and thus $L_{\mathrm{thinnest}} \approx 0.075m$.

### C1.2    Response time of sensors

The overturns that can be detected also depend on the response time of the sensors. The response time of the temperature sensor is $t_\theta^{\mathrm{response}} = 0.065$s (Seabird Scientific, 2025) and can be used to indicate over what vertical length scale the measurements are obtained (Galbraith and Kelley, 1996). The response time is recalculated into a vertical length scale $L^{\mathrm{response}}$ using:

$$L_\Theta^{\mathrm{response}} = \max [t_\Theta^{\mathrm{response}} v_{\mathrm{vmp}}(z)] \approx 0.06m. \tag{C2}$$

Note that $L_\Theta^{\mathrm{response}}$ changes depending on the vertical speed of the VMP, which may differ somewhat per profile depending on stratification and the exact mass of the weights added on. However, based on $L_{\mathrm{thinnes}}, L_\Theta^{\mathrm{response}}$ we choose a vertical averaging scale of $\delta_z = 0.08$ m.

### C2    Defining overturns and overturn-averaged values

### C2.1    Defining overturns

The raw CTD data is bin-averaged to the averaging scale as defined above. Vertical overturns are defined based on this measured profile $\Theta(z)$ and on $\Theta_{sorted}$. Here $\Theta_{sorted}$ is the sorted $\Theta(z)$ in descending order, representing a stably stratified background condition. For each $\Theta(z)$ one can find its exact equivalent in the profile $\Theta_{sorted}(z_{sorted})$. The vertical displacement of each sample is given by the vertical difference between a value of $\Theta$ in both the measured and sorted profiles:

$$\delta z = z_{sorted} - z. \tag{C3}$$

This gives a profile of the vertical displacement of each sample. Overturns are defined by the zero crossings of the vertical integral of this displacement profile (Thorpe, 1977; Mater et al., 2015). For each overturn, the Thorpe length scale is given as the root-mean-square of the vertical displacements $\Delta z$ of all samples within that overturn:

$$L_T(l) = \sqrt{\frac{1}{M_l} \sum_{m=1}^{M_l} (\delta z_m)^2}. \tag{C4}$$

Where $l$ indicated an overturn and $m = 1 : M_l$ the index of the samples within the overturn. $M_l$ thus depends on the size of the overturn and can vary between overturns.

The Ozmidov scale $L_{Oz}(l)$ and dissipation rate $\varepsilon_{TP}(l)$ per overturn are calculated using the overturn-averaged values of $N^2$ and $\varepsilon_{\mathrm{vmp}}$. Both $N^2$ and $\varepsilon_{\mathrm{vmp}}$ are first interpolated to the averaging scale of the CTD data, before being averaged for each overturn. The same procedure is followed for calculating the stability ratio per overturn.

## C2.2 Calculating overturn-averaged variables

To calculate $N^2$ we first average the CTD data of $\Theta$ and $S_A$ and $p$ into 1m vertical bins. This generally already averages over about 80% of the overturns (not shown). Subsequently, we stabilize the profile over 50m segments using the method from Barker and McDougall (2017) embedded in the TEOS-10 GSW-software (McDougall and Barker, 2011; IOC et al., 2010). We then interpolate the stabilized $N^2$ to the original vertical grid (of 8cm) and only use the middle 30m of the 50m section. This is repeated to cover the whole vertical profile. Using the 1m-binned data, the vertical gradients of $\Theta$ and $S_A$ are calculated and subsequently interpolated onto the 8cm grid. These are then used to calculate the stabillity ratio and averaged over an overturn.

## C3 The measurement accuracy used for criterion 2

The accuracy or precision of the measurements have an influence on the vertical scale over which measurements are significantly distinguishable, and thus a minimal vertical length overturns need to have. Using the precision of $t_\delta = 5 \times 10^{-5}$K for SBE3 temperature sensor (Seabird Scientific, 2025), we find

$$\delta_\Theta = \frac{\max\left[\Theta\left(S_A, t + t_\delta, p\right) - \Theta\left(S_A, t - t_\delta, p\right)\right]}{\sqrt{N_{O,l}}}. \tag{C5}$$

Here $\delta_\Theta$ is the error estimate on each $\Theta$ measurement over an overturn. We divide this error by the background gradients to obtain a vertical length scale which one must exceed to be able to distinguish one measurement from the other:

$$L_\Theta^{\text{error}} = \delta_\Theta \times \left(\overline{\frac{\partial \Theta}{\partial z}}^l\right)^{-1}. \tag{C6}$$

The resulting criteria mean that the size of the overturns needs to be larger than the length scale associated with this error, which is $\mathcal{O}(10^{-4} - 10^{-2})$ m (Fig. C1).

## C4 Additional figures

Based on the criteria defined in Sec. 5.2, part of the detected overturns are deemed unsuitable for estimating $\varepsilon_{\text{TP}}$. The effect of the criteria is shown in Fig. C1. Here station NP1 is taken as an example, but other stations show similar patterns.

Table 2 discussed the accuracy of the methods 1-4. In Fig. C2 a cross-validation of the estimates against $\varepsilon_{\text{vmp}}$ are shown in support of Tab. 2.

Figure 6 showed the estimates of $\varepsilon_{\text{TP}}$ for all four of the discussed methods, but with the estimates binned in halfoverlapping 200m bins for readability. The estimates at the 25 m resolution as was used in the analysis, is shown for three example stations in Fig. C3.

## Appendix D: The Triple Decomposition framework

Here we will discuss the tracer variance budget and how background profiles are obtained from an individual profile.

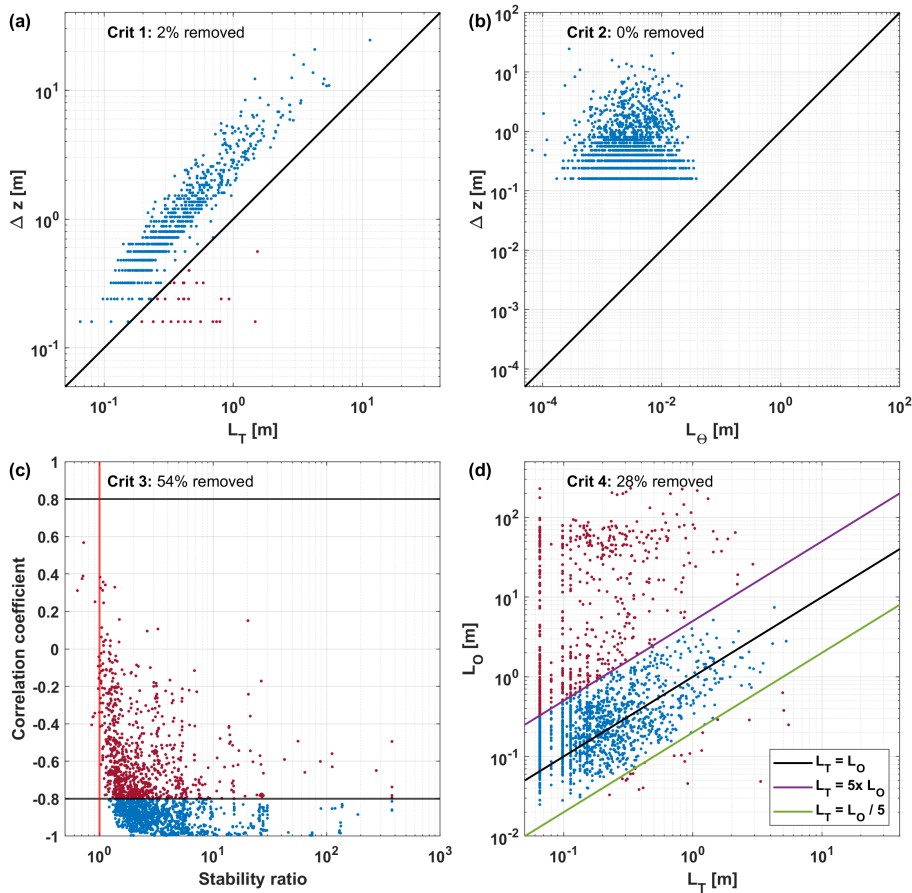

**Figure C1.** The overturn criteria applied on all detected overturns of a station. (a) Criterion 1: Thorpe length vs overturn size, (b) Criterion 2: measurement accuracy vs overturn size, (c) Criterion 3: stability ratio and correlation coefficient and (d) Criterion 4: relationship between Thorpe and Ozmidov scales. Red dots are flagged and will be removed from further calculations.

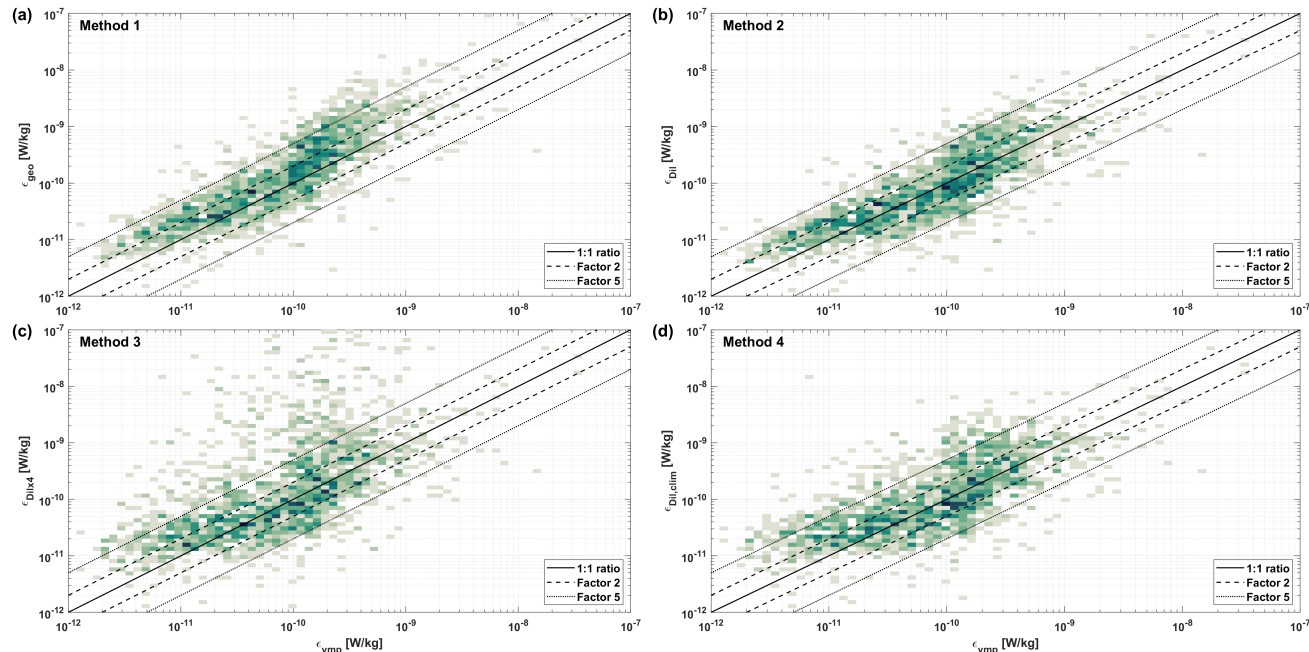

**Figure C2.** Cross-validation of the different methods, as defined in Section 5.4, against the corresponding values of $\varepsilon_{\mathrm{vmp}}$.

Assuming stationarity, homogeneity and non-divergence, the simplified tracer variance budgets, connecting the large-scale to the mesoscale and the mesoscale to the microscale, become (Garrett, 2001; Ferrari and Polzin, 2005)

$$\langle \mathbf{u}^e \Theta^e \rangle \cdot \nabla \Theta^m - \langle \widetilde{\mathbf{u}^t \Theta^t} \cdot \nabla \Theta^e \rangle = 0 \tag{D1a}$$

$$\langle \mathbf{u}^t \Theta^t \rangle \cdot \nabla \Theta^m + \langle \widetilde{\mathbf{u}^t \Theta^t} \cdot \nabla \Theta^e \rangle = -\frac{1}{2} \langle \chi \rangle. \tag{D1b}$$

In the context of the triple decomposition framework, we will use angled brackets $\langle \cdot \rangle$ to denote an average over a scale large compared to the scales within the brackets, but small compared to one scale larger in the framework. Summing these two equations results in,

$$\underbrace{2\langle \mathbf{u}^t \Theta^t \rangle \cdot \nabla_\perp \Theta^m}_{P_{\Theta^2}^\perp} + \underbrace{2\langle \mathbf{u}^e \Theta^e \rangle \cdot \nabla_\parallel \Theta^m}_{P_{\Theta^2}^\parallel} = -\chi_\Theta. \tag{D2}$$

Here $\nabla_\perp$ and $\nabla_\parallel$ are the across and along isopycnal gradient operators. Eq. (D2) states that the sum of the variance production on the microscale ($P_{\Theta^2}^\perp$) and the variance production on the mesoscale ($P_{\Theta^2}^\parallel$) are balanced by the variance dissipation ($\chi_\Theta$) as measured by the microstructure profiler. Using a gradient-flux approximation (Osborn and Cox, 1972), the production terms of Eq. (D2) can be written in terms of the isoneutral and dianeutral diffusivities;

$$2K|\nabla_\parallel \Theta^m|^2 + 2D|\nabla_\perp \Theta^m|^2 = \chi_\Theta. \tag{D3}$$

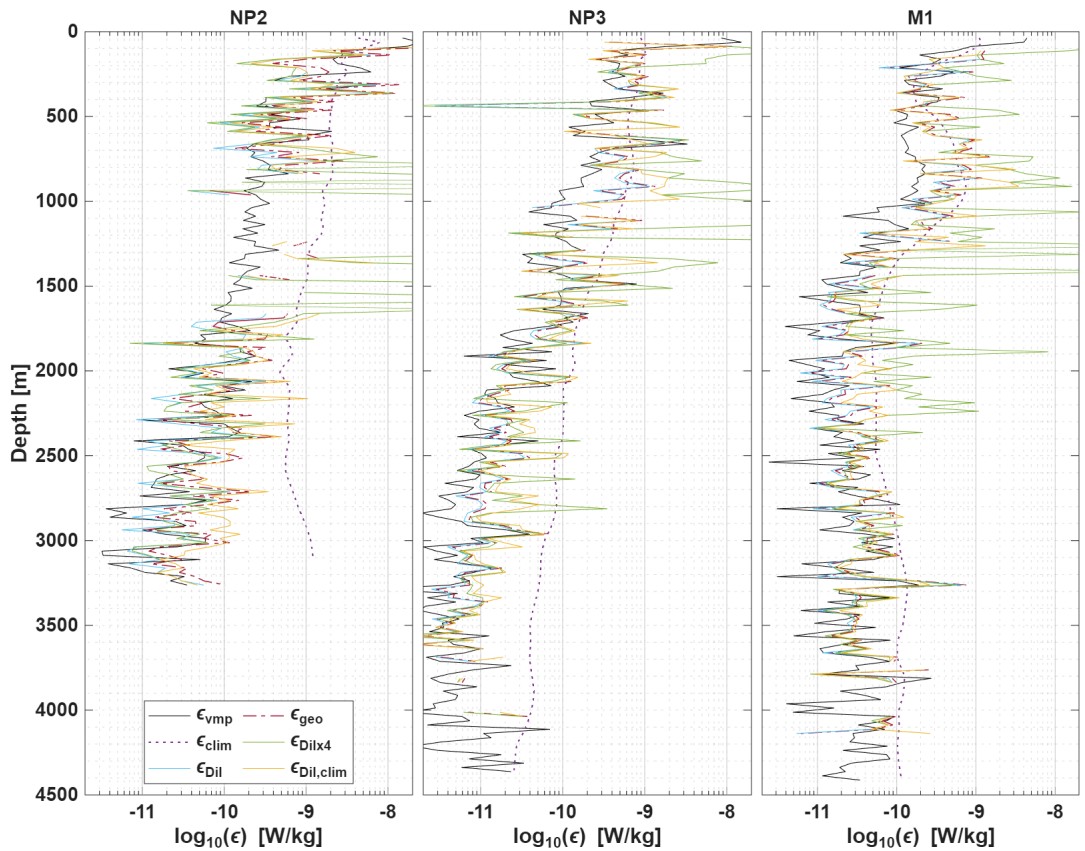

**Figure C3.** Dissipation estimates $\varepsilon_{\mathrm{TP}}$ for all different methods ($\varepsilon_{\mathrm{geo}}$: Method 1, $\varepsilon_{\mathrm{Dil}}$: Method 2, $\varepsilon_{\mathrm{Dilx4}}$: Method 3 and $\varepsilon_{\mathrm{Dil,clim}}$: Method 4, as defined in Sec. 5.4) are compared to $\varepsilon_{\mathrm{vmp}}$ and the used climatological estimates $\varepsilon_{\mathrm{clim}}$, shown for three example stations.

Where $K$ is the isoneutral diffusivity and $D$ the dianeutral diffusivity. This approach assumed that dianeutral tracer gradients are much larger than isoneutral gradients and that the isoneutral slopes are approximately flat.

### D1 Assessing background gradients from CTD data

In order to calculate the dianeutral production term, it is needed to obtain dianeutral gradients of the background scale temperature distribution from single CTD profiles. For this, a rationale similar to Castro et al. (2024) is followed. As the profiles are
755 too long to have an appropriate fit over the entire profile, a fourth order polynomial is fitted to $\overline{\Theta}(z)$ over a 200dbar window. At the window midpoints, the vertical derivative of the of fitted polynomal is taken and stored for the background profile. For the upper and lower ends of the profile, within half a window size of boundary, the derivative of the first (cq. last) fitted polynomal is taken.

## D2 Isoneutral diffusivities from the triple decomposition framework

The triple decomposition framework can be leveraged to obtain a first order estimate of the isoneutral diffusivity $K$ ($\mathrm{m^2\,s^{-1}}$) from the microstructure data. By assuming a flux-gradient approximation, the isopycnal production term $P_{\Theta^2}^{\parallel}$ can be written as,

$$P_{\Theta^2}^{\parallel} = 2\langle \mathbf{u}^e \Theta^e \rangle \cdot \nabla_{\parallel}\Theta^m = -2K|\nabla_{\parallel}\Theta^m|^2. \tag{D4}$$

Where $\nabla_{\parallel}$ indicate the along isopycnal tracer gradients. Combining Eq. D2 with the Eq. (12), for the dianeutral diffusivity and the expression for the isoneutral production above, results in an expression for the isoneutral diffusivity $K$ ($\mathrm{m^2\,s^{-1}}$);

$$K = \frac{\chi_\Theta - 2D|\nabla_{\perp}\Theta^m|^2}{2|\nabla_{\parallel}\Theta^m|^2}. \tag{D5}$$

For the isoneutral tracer gradients ($\nabla_{\parallel}\Theta^m$) we apply the VENM method of Groeskamp et al. (2019) to World Ocean Atlas 2018 1° resolution hydrological dataset (WOA18, Locarnini et al. (2019); Zweng et al. (2019)), providing the appropriate resolution for the isoneutral background gradients. The difference term, $\chi_\Theta - 2D|\nabla_{\perp}\Theta^m|^2$, is calculated at CTD-resolution, before being interpolated to WOA18 depths and divided by the isoneutral gradients over the range available (which is not always full depth, e.g. st. M2 in Fig. D1). When $\chi_\Theta/2$ and $D|\nabla_{\perp}\Theta^m|^2$ are of similar magnitude (at CTD resolution), the latter can occasionally be larger than $\chi/2$. This would lead to a negative $K$, such that we set the minimum to 0. After repeating this exercise for salinity data, this results in two sets of mesoscale diffusivity estimates $K$ (Fig. D1).

Most estimates of $K$ are centered between $\mathcal{O}(10^2 - 10^4)$, but there is a large degree of variation over 5 orders magnitude. Hence, the resulting $K$ can at best be considered a first order estimate of $K$, and are of similar magnitude and variabillity as estimated by Spingys et al. (2021) and Naveira Garabato et al. (2016) using a similar method. We find that $K$ is surface-intensified and roughly follows the same pattern as the climatological estimates of Groeskamp et al. (2020) in the upper 2000 dbar. Below that, the two estimates deviate substantially. Although the estimates of Groeskamp et al. (2020) are based on the first surface modes that are forced to decay to zero at the bottom by construction (LaCasce and Groeskamp, 2020), the large estimates of $K$ observed below 2500 dbar (especially St. NP4-NP8) are, when compared to previous diffusivity estimates (see e.g. Abernathey et al. (2021) Tab. 9.1, and references therein), very high. In fact, they may result from (very) small isoneutral background temperature gradients. This is emphasized by the empirical threshold of $|\nabla_{\parallel}\Theta^m| \leq 10^{-7}$ K m$^{-1}$, marked by the gray shading in Fig. D1, providing a rough estimate (based on Groeskamp et al. (2019)) where the estimated gradients are very small and subject to noise such that estimates of $K$ are easily overestimated. Although some previous studies suggest a reasonable comparison between the isoneutral production and the sum of the dianeutral production and dissipation terms (Cherian et al., 2024), we here conclude that these estimates of the isoneutral diffusivity $K$ are not very accurate. They show variations over multiple orders of magnitude, with extremely enhanced values in the deep ocean due to small isoneutral gradients. Even in the upper part of the water column, where these estimates seem to be of some value, they vary over many orders of magnitude of small ranges of depth, possibly indicating some limitation of the triple decomposition method to estimate mesoscale diffusivites. The noise in the estimates can in part be caused by the fact that these are estimates based on

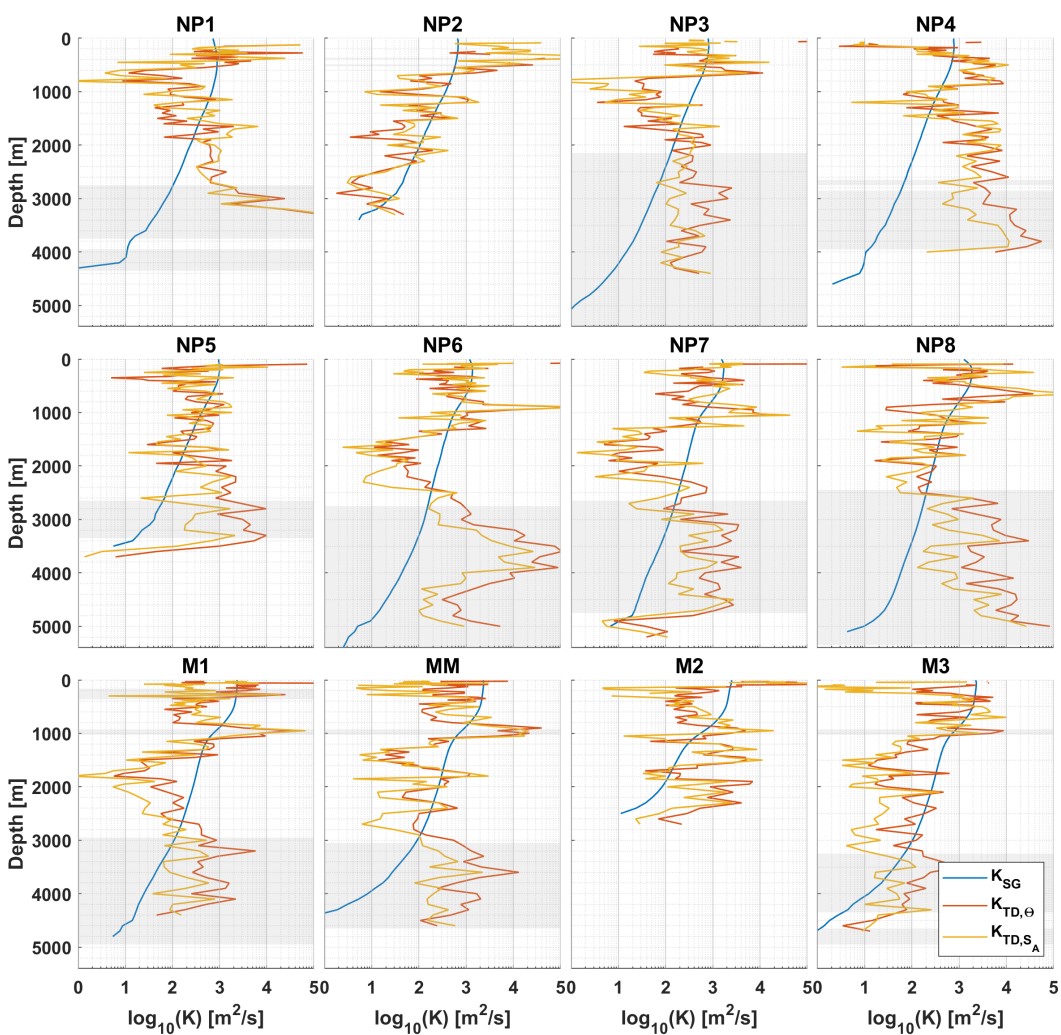

**Figure D1.** Estimated isopycnal diffusivities using the triple decomposition framework, based on both temperature and salinity data. Estimates are compared to climatological estimates of $K_{\mathrm{SG}}$ of Groeskamp et al. (2020). Depths for which $|\nabla_{\parallel}\Theta^m| \leq 10^{-7}$ are shaded in gray.

single profiles. It would help to average the estimates over appropriate time and length scales that fit the mixing processes associated with the isoneutral diffusivity. We conclude that the reliability of isoneutral diffusion estimates may be limited in regions of limited stirring signal, such as the ones sampled in this study. A more thorough quantification of signal-to-noise levels in the decomposition of variance dissipation rates, and in the estimation of isoneutral gradients, would be required to fully assess the reliability of the derived isoneutral diffusivities, which is beyond the scope of this study. However, the method is likely to produce better results in areas with stronger tracer gradients, where the finescale stirring signal is more prominent, such as the Antarctic Circumpolar Current (Naveira Garabato et al., 2016; Merrifield et al., 2016; Orúe-Echevarría et al., 2023) or subpolar regions (Castro et al., 2024).

*Acknowledgements.* We thank the captains and crew of the R/V Pelagia and R/V Atlantic Explorer for their assistance during the data collection. We also thank A. ten Doeschate and E. Cervelli for their helpful comments and support with the microstructure measurements and the processing of the data. Furthermore, we thank C. Whalen for useful discussion on the finescale parameterization.

*Code and data availability.* The raw and processed microstructure and CTD data belonging to the Mixation project (Stations M1-M3,MM in this manuscript) are available as Kusters (2025a). The microstructure and CTD data belonging to the Nanoplastics 2 project (Stations 1-8 in this manuscript) are available as Kusters (2025b) For the processing scripts of the shear data, the interested reader is referred to Rockland Scientific International Inc. Processing scripts for the thermistor data are available via Piccolroaz et al. (2021). World Ocean Atlas 2018 data can be downloaded from https://www.ncei.noaa.gov/access/worldocean-atlas-2018/. The VENM Matlab code of Groeskamp et al. (2019) is available at https://github.com/Sjoerdgr/VENM. Observational estimates of the passive tracer diffusity from Groeskamp et al. (2020) and related Matlab scripts are available as Groeskamp (2020). The climatological dissipation estimates of de Lavergne et al. (2020) are available as de Lavergne (2020).

*Author contributions.* NK: Conceptualization, Formal analysis, Methodology, Visualization, Writing - original draft. SG: Conceptualization, Methodology, Writing - review and editing, Funding acquisition, Supervision. BFC: Methodology, Writing - review and editing. HvH: Methodology, Writing - review and editing.

*Financial support.* This publication is supported by the project "Measuring the immeasurable: mapping of ocean mixing" with project number OCENW.M20.196 of the research program Open Competition Domain Science (ENW) which is financed by the Dutch Research Council (NWO). This publication is supported by the project "The intermittency of large-scale ocean mixing" with project number 2020.3 financed by the Utrecht University and NIOZ research collaboration fund. B. F. C. is supported by an European Research Council – Consolidator Grant 101169952 (REMIX-TUNE), and by an Advanced Research and Invention Agency – Forecasting Tipping Points grant SCOP-PR01-P021 (POLEMIX).

*Competing interests.* One of the (co-)authors is a member of the editorial board of Ocean Science (OS).

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
