# Peer review of "Microstructure Observations and Mixing Parameterizations along an Atlantic Transect in Very Weak Turbulence"

_EGUsphere, 2025_

## Author Comment (AC1)

**Review of Referee #1**

Summary:

This study demonstrates the observed turbulent energy dissipation rate along a transect in the North Atlantic, which includes the microstructure observations of the deep ocean turbulence. The authors compare the microstructure measurements with the finescale parameterizations and the Thorpe sorting method, which can be estimated using the standard CTD data. Thought these comparisons, they suggest some recommendations and restrictions for applying both parameterizations in a deep ocean weakly turbulent environment. While the paper is well-motivated and contributes to the science, I believe the manuscript requires substantial revision before it can be considered for publication.

We thank the referee for taking the time to read our manuscript and providing constructive feedback. Please find the original review in black, and our response and proposed changes added in blue.

Major comments:

I found this paper is concise and readable and the methods on the whole easy to follow. The comparison between the microstructure measurement and parameterization in the deep ocean is a good contribution to the existing literature, and most of the method is sound and adequately explained. However, for readers slightly removed from this field, I think that it would be useful to provide a further discussion in the Introduction about previous comparisons of finescale parameterizations and Thorpe methods to direct microstructure measurements.

Thank you for the suggestion. We have extended the discussion of previous comparisons in lines 61-65 with " *i.e. locations where sensor limitations are of lesser concern. Few studies do make a comparison between direct measurements of ε and one or more parameterizations in a low energetic environment, such as the Arctic Ocean (Fine2021,Baumann2023). However, these studies often only use shear-based observations of ε and are thus unable to consider values below the shear noise floor. Whereas for the few studies that are concerned with thermistor-based ε estimates in the deep sea (e.g., Scheifele2018,Yasuda2021), a comparison with parameterizations is not within the scope of those studies.*"

Section 8 of the current manuscript is mostly a summary of the results, but I think more discussion should be provided on the reasons why the finescale parameterization and Thorpe scale method deviates from microstructure observations in some regions of the present study, based on a discussion in the Introduction about previous comparisons. For

example, the overestimations of the parameterization shown in Figure 5 and Figure 6 are currently interpreted as being due to iso-neutral stirring, but the influence of other physical mechanisms (e.g., double diffusion) should also be discussed.

We have added a T/S diagram of the profiles as Fig. B1. Most of the data is susceptible to double diffusion (R_rho>1), but in the individual profiles of T and S (not shown) there are no consistent thermohaline staircases visible. However, the T/S diagram shows interleaving variability in most of the profiles where also the triple decomposition is showing an increased signal of isopycnal stirring. This supports the idea that isoneutral stirring is at least dominating over double diffusive processes. This discussion has been added in lines 473-476.

It has also been pointed out by previous studies that there the limitations of the finescale parameterization due to the fixed R_omega

Yes, we acknowledge that there are limitations to the fixed R_omega and that a variable R_omega could lead to better results: '*It can be argued that the best choice to gain optimal results would be to use a variable R_omega (Sun et al, 2024)'* (Lines 214-215)

and that the low-resolution of the CTD may not capture small overturns leading to the overestimation of Thorpe-method (Sheehan et al., 2023).

Thank you for suggesting this reference. It has been added in line 379 and lines 549-554 of the revised manuscript

After the revision of the introduction and discussion part, I believe that this paper will more accurately provide the science significance of this study in the context of other literature.

My other comments are mostly on details that could be improved.

Minor comments:

Line 35: Units used in parentheses should not be italicized here.
Corrected

Line 39: 'temperature variance'. Consider replacing it with "thermal variance dissipation rate'/'the rate of decrease of thermal variance' etc.
Replaced with 'thermal variance dissipation rate'

Line 54: 'fine-scale'but 'finescale' in the other parts. Please check for consistent spelling.
'Fine-scale' has been replaced for 'finescale' in lines 54, 63, and 470 of the revised manuscript

Line 71: 'strain rates'. Please rephrase it 'strain variance' or 'strain'.
Rephrased as 'strain'

Line 97: Estimates of εµU (εµT) were yielded from one distinct shear (FP07) probe? If they were obtained from multiple probes, are they averaged between them?
Yes, unless there was a large discrepancy between the two simultaneous estimates, the two estimates are averaged. This is described in line 626 of the revised manuscript, together with the other quality control metrics.

Section 3: Showing some example spectra (or composite spectra) for the shear and FP07 method in the appendix would be very helpful. Plotting the observed spectra with the fitted theoretical curves and some characteristics wavenumber (e.g., Batchelor wavenumber) would also make it easier for readers to follow the explanation of the methods for the estimates of εµU and εµT .
Thank you for the suggestion. Example spectra have been added as Figure A1.

[Figure]

Line 125: (Fer et al. (2014)) - > (Fer et al., 2014)
Corrected

Figure 3: typo in the legend of Figure 3b.' perrcentile' -> ' percentile'. Also, plotting the lines of shear probe noise floors in Figs. 3a, c as well as Figs. 3b, d would be helpful.
The typo has been corrected. Also the lines showing the different methods for defining the noise floor, as was shown in Tab 1, have been added to panels 3a,c.

Table 1: Why the values of the mode (Scheifele et al., 2018) is smaller than the 5th percentiles (Piccolroaz et al., 2021). Please confirm it.
Thank you for spotting this, indeed the values have been swapped. This has been corrected.

Line 155: The citations should be inside the parentheses.
Corrected

Line 156: Please include references.
A reference to Piccolroaz et al (2021) has been added for the factor of 1.3 and a reference to Technical Note 40 of RSI has been added for the noise curve.

Line 157: What is 'x %' in this sentence?
Thank you for spotting this. The 'x%' has been replaced by '0.49%'

Line 160: Was the correction method to compensate for the insufficient temporal response (Lueck et al. 1977; Gregg and Meagher 1980; Oakey 1982; Gregg 1999) used in this study? Did you use the same correction method as Piccolroaz et al. (2021)? This should be clearly stated in the main manuscript.
The time response has been corrected using a double-pole transfer function. The first part of line 168 has been rewritten to: "Whilst the limited, speed-dependent time-response is corrected for using a double-pole transfer function (Vachon and Lueck, 1984), ..."

Line 206: (Waterman et al. (2013); Chin et al. (2016); Fine et al. (2021)) - > (Waterman et al., 2013; Chin et al., 2016; Fine et al., 2021)
Corrected

Line 213: I think that this sentence is not clear. How can we interpret the relationship between R_omega and the depths from Fig. 4? Please consider rephrasing this sentence (or revising Fig. 4).
We have removed this sentence.

Line 221: How did you define 'the surface mixed layer' here? Did you just omit the several bins close to the surface?
Yes, binning of the data for the finescale method was started at 100 dbar. 'the surface mixed layer' has been rephrased to 'the upper 100 dbar' to reflect this. We have also noted that the surface mixed layer was shallower than 100 dbar for all profiles.

Line 229: I think that it would be very helpful to plot vertical profiles of temperature, salinity, and R_rho and/or T-S diagrams to show the water mass variability in the present study. These plotting would be useful when you discuss how the region, where parameterizations were over(under)estimated, correspond to the finescale watermass variabilities. Also, when interpreting the results hereafter, it is important to indicate whether the water column is susceptible to double diffusion or not.
Thank you for the suggestion. A TS diagram has been added as Fig B1 and the following text has been added in lines 235-237: "*Finestructure watermass variability in the form of interleaving patterns are observed in the T/S diagram (Fig. B1). The depths where these*

*patterns are observed correspond to the locations where temperature-based finescale estimates (severely) overestimate the measured dissipation rates ε_{vmp}."*

Line 234: Why does salinity noise at weak stratification lead to the 'underestimation' of parameterization? Please explain it in more detail.

At this point we are not fully certain of the exact cause of the observed underestimation. The spectra of these segments are only minimally bluer than the segments that do agree well with the microstructure data, so that does not fully explain the observed underestimates. Also the next paragraph (l. 249-252), gives a possible explanation based on the shear-to-strain ratio, but unfortunately that is difficult to test with the absence of shear data (by calculating R). However, at this point flagging and removing these segments based on the proposed criteria seems to work well.

Line 305: App.D3 is not found in the manuscript.

This should have been App. C3 and has been corrected.

Line 326: I'm confused by this condition. As far as I can tell, the condition R_rho>1 corresponds to the salt-fingering-favorable regime (e.g., Schmitt, 1994). Why did you not include R_rho<-1 (doubly-stable regime). Under the condition R_rho>1, I'm afraid that the temperature change would capture the structure of the staircases and intrusions due to double diffusion rather than mechanical turbulence. Please explain this part in more detail.

Thank you for raising this point. The intention of the R criterion is to filter out overturns where temperature is not the dominate factor (-1<R<1). You are right that it should include R<-1 as well. We changed the mention of R>1 in l.330 and in Eq 6 to be |R|>1, so that it also accounts for R<-1. We did not encounter this in our calculations, so adding the absolute value did not alter the results.

Line 358: Did the vertical size of the overturn larger than 25m exist?

Only in a few instances. The results are calculated per overturn and given on the mid-point depth of the overturn. The mid-point depth is used for the averaging, so there is no double counting of overturns/dissipation rates in the averaging process.

Line 377: Since the overestimation and underestimation were considered together in Table 2, it would be helpful to have scatterplots (eps_TP versus eps_VPM) to look at overestimation and underestimation separately.

These plots have been added as Fig C2.

[Figure]

Subsection 5.5.2: As shown in Fig. 6, the criterion 4 mainly works as a rejection of overestimate beyond a factor of 10 (When assuming the criterion 4 leads to the removal of the overturn for which eps_TP< 1/25 eps_VMP and eps_TP > 25 eps_VMP ). What is the cause of these overestimations?

We are not quite sure. Overestimations are also more abundant than underestimations it seems. This suggests that the Thorpe scales are so large that they lead to overestimation of dissipation compared to the VMP measurements. And as these overturns are otherwise removed by the Ozmidov scale criterion, this suggests that the turbulence might not be isotropic. Further study is needed to better understand this phenomenon. For now, criterion 4 seems to do the job to account for this.

Line 385: Does St.2 and 3 in this line refer to St. NP2 and NP3?
Yes, this has been corrected.
How can we interpret from Fig.2 that 'parameterized estimate ($\varepsilon$TP?)' tends to be more similar to  than ? Please consider rephrasing it.
The last part of this sentence has been removed.

Line 393: 'about 30-40% is removed' by which method among Method 1-4?
This is mainly by methods 3, and 4 if this is applied. The text '(mostly by criterion 3 and 4 if applied)' is added in line 403-404.

Line 453: What did you use as D in calculating /in Fig7? Is it the same as in Eq.12? If so, please define here.
Yes, a reference to Eq 12 has been placed at the end of line 455.

Line 462: I think that it is necessary to discuss the possibility that the high values of  is affected by the double diffusion rather than the isopycnal stirring. I understand that it is difficult to conclude it, but it would be useful to discuss various possible mechanisms other than the mechanical turbulence.

See our response to the next comment.

Line 465: I understand that it is difficult to see horizontal T/S variations from the observations due to the sparce locations, but I think it would be useful to show the vertical distribution of temperature, salinity, and Rrho. Is there any interleaving patterns in temperature and/or salinity in the vertical profiles of T,S?

It is indeed difficult to assess horizontal variations with the current dataset. We have added a T/S diagram of the profiles as Fig. B1. Most of the data is susceptible to double diffusion (R_rho>1), but in the individual profiles of T and S (not shown) there are no consistent thermohaline staircases visible. However, the T/S diagram shows interleaving variability in most of the profiles where also the triple decomposition is showing an increased signal of isopycnal stirring. This supports the idea that isoneutral stirring is at least dominating over double diffusive processes. A comment discussing this has been added in l473-477.

Line 510: Does 'St. 4-8' mean St. NP4-8?

Yes, it does, it has been corrected.

Line 542: 'temperature-based'

Corrected

---

## Author Comment (AC2)

**Review of Referee #2**

The authors provide a very detailed analysis of a series of microstructure profiles collected in the North Atlantic. They provide a detailed breakdown of the data and technical analysis of the performance of the instrument and comparison against non-microstructure approaches to estimating turbulent parameters. Through this they provide advice on the applicability and analysis of both microstructure and CTD data for turbulence. This includes considering the limitations of both fine scale parameterizations and Thorpe scale analysis which the authors interpret as being driven by the methods not accounting for isoneutral mixing processes. In general, this is a very detailed manuscript which is well written and provides interesting insight. The discussion nicely summarises the results and recommendations from this study however it would be much stronger if there was a discussion of how the recommendations made here compare with the results of other studies.

We thank the referee for taking the time to read our manuscript and providing constructive feedback. Please find the original review in black, and our response and proposed changes added in blue.
We have added extra discussion on how the results compare with other studies in the Introduction (l.61-65 and in section 8, Discussion & Conclusions (l. 533-538, l.548-554)

Detailed comments

Line 72 – space missing between deep and sea
Space added

Line 157 – "Only x% of" missing value
Thank you for spotting this. The 'x%' has been replaced with '0.49%'

Line 193 – I suggest to clarify what you mean by shear as you did collect microstructure shear data so I assume you mean LADCP based shear. I suggest "shear" > "large-scale shear"
Thank you for the suggestion. 'shear' has been replaced for 'large-scale shear'

Around line 275 – Is it true that your CTD is measuring at 8cm vertical resolution? I would be concerned about the impact of the thermal mass of your platform and the flushing time of CTD. I saw you quote the response time given by SeaBird in the appendix however that is not the time for the sensor to give the "true" value. I think the more relevant value is the settling time which can be quite a bit longer. These potential constraints should at least be mentioned.

The CTD is sampling at a rate of 64Hz (approx. 0.012m resolution, depending on fall speed), which is effectively a higher resolution than 0.08m (depending on fall speed). As you already indicate, there are several factors that influence response length scale, such as the sampling frequency, profiler fall speed and sensor response times. This is extensively described in Appendix C1.2. From this, it follows a typical value of 0.06m. Because of variations in this value, the more conservative estimate of 0.08m is chosen. Additionally, we note that the CTD instrumentation was mounted on the free-falling VMP. The thermal mass of this platform is not of concern, as the profiler is monotonously falling and thus not affected by e.g. ship-heave as would be the case for a standard ship-tethered CTD-rosette.

Line 349 – I think the full stop is meant to be a comma, as what follows isn't a sentence
This sentence was complicated, it has been rewritten completely.

Line 366 – I think the labels for alpha for each method are the wrong way round
Thank you for spotting this

Line 510 – The statement that the values are "likely unrealistically high" should be justified or removed. Whilst the authors might believe this, if it is to appear in a published paper it need to be driven by either data or some other evidence e.g. a scaling that shows that such values are not consistent with our existing understanding of the deep ocean. The authors compare with the profiles of Groeskamp et al 2020 however that approach also contains many assumptions and it is not clear to me a priori that one approach is more robust than the other.
The part of the isoneutral diffusivities has been moved to the Appendix D2, including most of this discussion. Nevertheless, in response to this comment, we agree with the reviewer that the method of Groeskamp et al 2020 is also based on many assumptions. This is clearly illustrated in the first part of the sentence. For further justification, we have added a reference to Abernathey et al 2022, Tab 9.1. For example, estimates from tracer experiments at large depths show very small isoneutral diffusivities, rather than the large values obtained in this study.
The sentence ending with "… are likely unrealistically high."  has been rephrased to "…are, when compared to previous diffusivity estimates (see e.g. Abernathey (2022), Tab. 9.1, and references therein), very high."

Line 613 –Typo lenght > length
Corrected

---

## Author Comment (AC3)

**Review of reviewer #3, Dr Deepak Cherian**

This paper is a fairly thorough look at microstructure measurements from a small number of profiles (approx 10) spanning a transect across the Atlantic Ocean. The amount of analysis done is impressive, and the methodological summaries are a useful resource for anyone new to the field. However, having read the paper twice now, I have some major comments; and recommend major revisions.

Thanks for reading the paper thoroughly. Your feedback is much appreciated and we have used it to improve the paper as we explain below. Our comments are in blue.

1. A lot of effort is spent in explaining the methods, so much so that it is hard to draw strong conclusions. Some of the shear-strain ratio analysis, and the less compelling Thorpe scale based estimates might be better suited to supplementary material. The discussion section needs some context on how these results might be novel; or if not, how they support an existing body of work in to weak turbulence regimes. In general, drawing strong conclusions from a small number of turbulence profiles is hard; so it might be useful to bring in previous datasets from the NSF microstructure database to provide extra context.

   We have added some context in the introduction how our study differs from existing comparisons (lines 61-65). Additionally, we have added more discussion placing our results in context of existing work in the Section 8, lines 533-538, and 548-554.

   We do acknowledge that the amount of data is relatively limited (lines 555-559) and that it is a caveat in our study. However, we don't think bringing in more datasets will change the conclusions and focus of this paper. The focus of this paper is 1) to compare thermistor data and shear data in low turbulence regions, 2) compare strain rate and Thorpe estimates directly to observed profiles, and 3) understand if the triple decomposition can give insights for interpretation. We stressed these points more in the introduction.

   These main results do not necessarily require more data and compiling a large set of turbulence measurements could be a study by itself. In addition, those datasets will either be in different locations, or years or even decades apart, and thus not likely to improve the results from this dataset, by averaging between profiles. We therefore want to stay with a comprehensible number of profiles so we can study individual differences. We also added the following note to lines 559-561: *"Adding more profiles will undoubtedly lead to better comparisons between the direct microstructure observations and the indirect parameterizations, but it seems unlikely that it will change the key conclusions of this paper."*

2. Much of the paper focuses on very low ε & χ values. It is not clear to me how the noise floor of the thermistor is being handled. At some point, there is no "overturning turbulence", and (to me) the only sensible thing to do is apply molecular diffusivities and set ε to 0. See Cherian et al (2020) for references (including Gregg et al 2012; Itsweire et al 1988). If you "discard" too many of these low turbulence segments, you will bias the mean. From the text it doesn't appear as if this is actually biasing the mean, but it would be good to be sure and to discuss the work of Itsweire and Gregg here.

The reviewer makes a few points that we would like to address separately.

First, the thermistor noise floor is discussed in lines 159-165 of the revised paper: *"In our processing routines we discard any data segment where the average spectral levels fall below a factor of 1.3 (Piccolroaz2021) of the well-characterised sensor noise curve (following RSI, see Technical Note 40 available at www.rocklandscientific.com). Only 0.49% of data-segments were discarded following that criterion. For the retained data, the distributions of ε_{\mu T} do not show a strong deviation from the expected log-normal distribution (Fig. 3a,c), as found for the shear sensor, suggesting that these estimates are still reliable and above the noise floor."*

So based on the low amount of segments being discarded following this criterion, plus the distributions seen in Fig 3, we argue that the noise floor of the thermistors is not reached for the data that is used. So it is unlikely biasing the mean by discarding too many segments, which would be the case if only the shear that is used.

Besides the handling of the data, we argue that the noise floor of the thermistors is not reached for the data that is used. At least, it doesn't show as a clear cutoff in the distribution of the data (Fig 3), while a cutoff is found for the shear data.

[Figure]

Second, we agree that at some point there is no overturning turbulence anymore. However, as can be seen in Fig 8, the diffusivities are still multiple orders above the molecular levels (smallest diffusivities are O(1e-6), but most of the time larger, whereas typical values of the molecular thermal diffusivity is O(1e-7)), and thus setting epsilon to zero does not make sense here.

A further analysis based on the buoyancy Reynolds number and the Cox number confirm this (see Figure). With only a minor part of the estimates having both the Reb and Cz below 10 (~6%). It is unlikely to significantly impact our results.

The dissipation rate and diffusivity would be biased high if we only used the shear probes (affected by noise floor). We believe that our approach of using a combined profile of shear and thermistor based dissipation estimates avoids the introduction of such bias, by using the most accurate and direct estimate available.

The "triple decomposition" analysis in Section 6 doesn't add much to the paper, and could be removed in my opinion. I completely agree with the conclusion (lines 510-520) that it is not useful over much of the data. But that is to be expected from an analysis method that relies on residuals. Combining that with uncertainties in estimating the very small isopycnal density gradient in the abyss, the resulting diffusivity values cannot be good. A formal uncertainty analysis should indicate that your residuals are indistinguishable from 0, but isn't worth the effort in my opinion. In my experience, it is quite hard to extract the signal even from very well sampled measurement campaigns like NATRE. Outside of spicy isopycnals, where there is very low isopyncal stirring, you will simply get junk if attempting to estimate a residual (this is easily visble in Ferrari and Polzin, 2005). If you'd really like to do the comparison, I'd recommend limiting yourself to regions where "spiciness" is large,

and avoiding diffusivity and comparing χ instead (if the χ residual proves to be significant).

We agree with the reviewer that this part of the manuscript can be improved. It is indeed indicated that the isopycnal diffusivity estimates don't have much value due to the highly variable results (l. 786-790). Therefore we moved this part about the isoneutral diffusivities to the appendices, so that an interested reader can still find it, but it is no longer part of the main text. We find it useful to keep these results to show that it is not useful in this situation and provide arguments why. However, we agree this does not fit the main results.

We have left Section 6 about the triple decomposition in place, as it supports the arguments made in other sections and the phenomena seen (l.469-486). These arguments pertain specifically to the areas where the triple decomposition does have a large signal, and thus the decomposition can be used.

Hopefully the reviewer can agree with this compromise.

---

## Author Response (AR2)

**Public justification (visible to the public if the article is accepted and published)**:
I see the authors addressed all comments from 3 reviewers mostly in an appropriate manner.

Thank you for pointing out these last few issues. Please find our response to each comment added in blue.

We also updated the contact information in response to the comments from the editorial staff.

The only place I find it awkward is at L.561: "it seems unlikely". The reason why the authors consider it "unlikely" can be found in the reply to Reviewer 3 (Comment 1), but not obvious in the revised manuscript. I understand the sentence is added in response to the comment, but I think the sentence might appear superfluous and in my opinion awkward.

If this phrase is to remain, the authors might add the reference to introduction to remind the scope of the work and/or add a comment to the effect that addition of more data with similar conditions is difficult.

Thank you for this comment. We have rephrased/expanded the sentence to *"Adding more profiles will undoubtedly lead to better comparisons between the direct microstructure observations and the indirect parameterizations, but it seems unlikely that it will change the key conclusions, as the focus of this paper is on individual differences between methods rather than the assimilation of a large quantity of microstructure profiles (see also Sec. 1)."*

L.167 "smoothest out" ?
Rephrased as "smooths out"
L.392 "de Lavergne et al. (2019, 2020)" → (de Lavergne et al, 2019, 2020)
Corrected
L.673 "(Piccolroaz et al., 2021; Ruddick., 2000)" → Piccolroaz et al. (2021) and Ruddick (2000)
Corrected

L. = Line numbers in the revised manuscript (egusphere-2025-3165-manuscript-version2.pdf)

Otherwise, the manuscript is almost ready for publication. Thank you for choosing Ocean Science.